# Examining the sensitivity of the terrestrial carbon cycle to the expression of El Niño

Lina Teckentrup[1,2], Martin G. De Kauwe[1,2,3], Andrew J. Pitman[1,2], and Benjamin Smith[4,5]

[1]ARC Centre of Excellence for Climate Extremes, Sydney, NSW, Australia
[2]Climate Change Research Centre, University of New South Wales, Sydney, NSW, Australia
[3]Evolution & Ecology Research Centre, University of New South Wales, Sydney, NSW 2052, Australia
[4]Hawkesbury Institute for the Environment, Western Sydney University, Penrith, NSW, Australia
[5]Department of Physical Geography and Ecosystem Science, Lund University, Lund, Sweden

**Correspondence:** Lina Teckentrup (l.teckentrup@unsw.edu.au)

**Abstract.** The El Niño-Southern Oscillation (ENSO) influences the global climate and the variability in the terrestrial carbon cycle on interannual timescales. Two different expressions of El Niño have recently been identified: (i) Central–Pacific (CP) and (ii) Eastern–Pacific (EP). Both types of El Niño are characterised by above average sea surface temperature anomalies in the respective locations. Studies exploring the impact of these expressions of El Niño on the carbon cycle have identified changes

in the amplitude of the concentration of interannual atmospheric carbon dioxide ($CO_2$) variability following increased tropical near surface air temperature and decreased precipitation. We employ the dynamic global vegetation model LPJ–GUESS within a synthetic experimental framework to examine the sensitivity and potential long term impacts of these two expressions of El Niño on the terrestrial carbon cycle. We manipulated the occurrence of CP and EP events in two climate reanalysis datasets during the later half of the 20th and early 21st century by replacing all EP with CP and separately all CP with EP El Niño events.

We found that the different expressions of El Niño affect interannual variability in the terrestrial carbon cycle. However, the effect on longer timescales was small for both climate reanalysis datasets. We conclude that capturing any future trends in the relative frequency of CP and EP El Niño events may not be critical for robust simulations of the terrestrial carbon cycle.

## 1 Introduction

The terrestrial carbon cycle varies markedly on interannual timescales and is significantly influenced by the El Niño Southern

Oscillation (ENSO) at global scales. Around 20% of the vegetated land shows a significant negative correlation with the ENSO cycles, predominantly in the tropics and in arid areas. Around 12% of vegetated land is positively correlated with ENSO cycles, with this correlation dominated by arid areas (Zhang et al., 2019). In general, ENSO is positively skewed such that El Niño events have a stronger effect on the terrestrial carbon cycle than La Niña events (e.g. Haverd et al., 2017; Ahlström et al., 2015). During El Niño events, terrestrial ecosystems typically act as a carbon source while during La Niña events, carbon

uptake is enhanced, particularly in semi-arid ecosystems (e.g. Ahlström et al., 2015). Multiple studies have examined the effect of El Niño on the terrestrial carbon cycle using observations and ecosystem models (e.g. Bastos et al., 2018; Rödenbeck et al., 2018; Zhang et al., 2019; Fang et al., 2017). Given the influence of ENSO on the interannual variability (IAV) of the terrestrial

carbon cycle, representing ENSO and associated teleconnections is important in coupled Earth system modelling (e.g. Kim et al., 2016; Qian et al., 2008).

Each El Niño event varies in terms of the pattern and intensity of sea surface temperature anomalies. Recent analyses have highlighted two distinct expressions or flavours of El Niño: the (i) Central-Pacific (CP) and (ii) Eastern-Pacific (EP) El Niño (Donguy and Dessier, 1983; Ashok et al., 2007; Weng et al., 2007). Both expressions of El Niño are characterised by above average sea surface temperature anomalies in their respective locations. Depending on the location of the sea surface temperature anomalies, these different expressions of El Niño are associated with different impacts on the Walker circulation,

different teleconnection patterns and therefore different regional-scale rainfall and temperature anomalies (e.g. Taschetto and England, 2009; Ashok et al., 2009; Weng et al., 2007; Ashok et al., 2007). For example, Taschetto and England (2009) found that maximum rainfall decreases associated with EP El Niño events tend to occur over northeastern and southeastern Australia while CP El Niño events are associated with a negative precipitation response in nortwestern and northern Australia. Further, the timing of the maximum precipitation anomalies varied with the expression of El Niño. Given the different expressions of El

Niño, and the consequential differences in regional rainfall and temperature, a different ecosystem response might be expected between CP and EP El Niño events. A shift in El Niño patterns could change cumulative net biome production, which may alter competitive patterns of plant functional types, both of which may influence the carbon stored in vegetation and soil (e.g. Park et al., 2020). Similarly, interannual variability in precipitation patterns induced by different types of El Niño might result in a shift in vegetation distributions, particular at climatic transition zones, or in water-limited environments, e.g. semi-arid

areas/savanna ecosystems (cf. Scheiter and Higgins, 2009; Whitley et al., 2017). Recent studies have found that depending on the expression of El Niño, different time lags, amplitudes and duration in the carbon cycle anomalies occur (Wang et al., 2018; Chylek et al., 2018). Wang et al. (2018) and Chylek et al. (2018) link the effects of different expressions of El Niño on the terrestrial carbon cycle to variability at interannual timescales but the impact on longer timescales is not well understood.

    While it is clear that El Niño has an impact on the terrestrial carbon cycle, analyses that have demonstrated this mostly have

not attempted to separate El Niño into CP and EP types. Understanding the sensitivity of the terrestrial carbon cycle to these distinct El Niño expressions is a key knowledge gap, specifically because there is evidence that the relative frequency of CP and EP El Niños may be changing. There is emerging evidence that in the late 20[th] and early 21[st] century the occurrence of CP El Niño events increased in frequency (Yu and Kim, 2013) and some studies using climate projections suggest that this trend will continue as the atmospheric carbon dioxide ($CO_2$) concentration increases (e.g. Yeh et al., 2009). Despite recent research

finding that the expression of El Niño is important at interannual timescales (Wang et al., 2018; Chylek et al., 2018; Pan et al., 2018), it is not known how and where the recent trend towards more CP El Niño events would impact the terrestrial carbon cycle.

    This re-focussing towards the specific expression of El Niño is potentially problematic for global climate models, which currently struggle to correctly resolve El Niño – La Niña cycles with the correct persistence and teleconnections (Bellenger

et al., 2014). If the expression of El Niño, as distinct from El Niño in general, is shown to affect IAV as well as the longer timescale terrestrial carbon balance, this would place a significantly higher demand on climate models to accurately reproduce

both the persistence and teleconnections of the El Niño – La Niña cycles, and the relative frequency in the future of CP and EP El Niño events. This could significantly constrain our capacity to predict the future of the terrestrial carbon cycle.

To explore whether the expression of El Niño affects the global and regional terrestrial carbon cycle on multi-decadal timescales, we use a dynamic global vegetation model (DGVM) forced by the climate data obtained from two reanalysis datasets. We generate two synthetic forcing data sets: one where, starting 1968, all El Niño events are a CP type and one where all El Niño events are an EP type. We then use our DGVM experiments to examine the impact of the expression of El Niño on the global terrestrial carbon cycle.

## 2 Methods

### 2.1 Model

LPJ–GUESS (Lund-Potsdam-Jena General Ecosystem Simulator; Smith et al., 2001, 2014) is a DGVM extensively used for climate-carbon studies (Smith et al., 2014; Sitch et al., 2003). LPJ–GUESS is used as the land surface scheme in the global Earth System Model, EC-Earth3 (Weiss et al., 2014; Alessandri et al., 2017) and in the regional Earth System Model RCA-GUESS (Wramneby et al., 2010; Zhang et al., 2014). LPJ–GUESS dynamically simulates the exchange of water, carbon and nitrogen through the soil-plant-atmosphere continuum (Smith et al., 2014), resolving the vegetation's resource competition for light and space. LPJ–GUESS groups the vegetation into 12 plant functional types (PFTs), simulating differences in growth form (grasses, broadleaved trees, deciduous trees), photosynthetic pathway (C3 or C4), phenology (evergreen, summergreen or raingreen), tree allometry and life history strategy, fire sensitivity and bioclimatic limits for establishment and survival.

We use LPJ–GUESS version 4.0.1 in 'cohort mode' where woody plants of the same size and age co-occurring in a local neighbourhood or 'patch' are represented by a single average individual. Each PFT is represented by multiple average individuals and one PFT cohort is defined as the average of several individuals. Assuming that all individuals of the same age in a particular patch have the same structure, then several cohorts form a single patch. Establishment, mortality and disturbance are stochastic processes. Fire is simulated annually (stochastically) based on temperature, fuel availability and the moisture content of upper soil layer as a proxy for litter moisture content (Thonicke et al., 2001).

### 2.2 Forcing

LPJ–GUESS requires soil texture (Zobler, 1986; Sitch et al., 2003), daily temperature, precipitation, incoming shortwave radiation and the annual mean atmospheric carbon dioxide ($CO_2$) concentration. The atmospheric $CO_2$ concentration, varying annually, was compiled from atmospheric measurements (McGuire et al., 2001; Smith et al., 2014). We used the CRUNCEP V7 dataset (Viovy, 2018) as the meteorological forcing input for LPJ–GUESS. CRUNCEP is based on a merged observed monthly climatology product of the Climate Research Unit (CRU) and the high temporal resolution reanalysis by the National Center for Environmental Prediction (NCEP). The spatial resolution is 0.5° and the temporal resolution is 6–hours for the

time period 1901–2016. From the CRUNCEP data, we calculated daily averages of the temperature and incoming shortwave radiation and daily sums of the precipitation as inputs for LPJ–GUESS.

To explore the sensitivity of our results to differences in the meteorological forcing, we repeated our experiments using the Global Soil Wetness Project Phase 3 (GSWP3) dataset (Kim, 2017). GSWP3 is based on the 20th Century Reanalysis (20CR; Compo et al., 2011) which is dynamically downscaled from a global 2° resolution with a 3h temporal resolution into a T248 (∼0.5°) grid using a spectral nudging technique (Yoshimura and Kanamitsu, 2008) in a Global Spectral Model. Since GSWP3 only covers the time period of 1901–2010, we choose the CRUNCEP-based simulations for the main analysis and use the GSWP3-based simulations to determine if the meteorological forcing leads to major differences in our results.

## 2.3 Model set–up

LPJ–GUESS was spun up for 500 years using the first 30 years of the climate forcing (1901–1930) to allow the carbon pools to reach equilibrium. During the spin-up, temperature is detrended and the climate forcing is cycled repeatedly with a constant atmospheric $CO_2$ concentration of 296 ppm. After the spin-up, the simulation continues with the historical climate and transient atmospheric $CO_2$ forcing (e.g. Smith et al., 2014). For this study, we allow for fire and stochastic disturbance. We do not account for recent anthropogenic changes in global land use cover.

## 2.4 Identification of El Niño events

We base the identification of El Niño events on a study by Yu and Kim (2013). They first classified El Niño events based on the Oceanic Niño Index (ONI) which comprise both CP and EP El Niño events. Based on four indices, they then further differentiate between CP and EP El Niño events. For this study, we define a CP or EP El Niño event when three out of these four indices agree on the same El Niño type. The remaining events are defined as mixed events ('MIX'; see appendix table A1; compare table 1 in  Yu and Kim, 2013).

Note that our approach defines the 1968–1969 El Niño event as the first CP El Niño event and consequently the first year of our experiment set-up. Given the climate forcing is limited to 1901–2015, we exclude the 2015/2016 El Niño event and choose the ENSO-neutral year 2013 as the final year. We analyse the effect that a climate with only CP El Niño or only EP El Niño events might have on terrestrial vegetation after 45 years by comparing the final year of the two different scenarios to that of the control run (where both expressions of El Niño occur).

## 2.5 Experiment design

In the control run, we ran LPJ–GUESS with the original CRUNCEP forcing for the period 1901–2015. For the experiment simulations, we created two climate forcing data sets containing either only CP or only EP El Niño events starting from 1968, hereafter referred to as: CP–only and EP–only. To do this, we replaced climate anomalies associated with CP El Niño events with those of EP El Niño events and vice versa for the three climate variables temperature, precipitation, and incoming

shortwave radiation. In this study, we focussed our analysis on the the tropics (23°S – 23°N) and Australia in addition to a global analysis.

To generate the synthetic CP/ EP forcing datasets, we take the reanalysis forcing (displayed schematically in fig. 1a) and first calculate eight 30–year–averages that are used as base periods (see fig. 1b) for every grid point based on the original climate forcing (see fig. 1a). Each base period is used to calculate the anomalies for successive five year periods, i.e. we compare the years from 1966–1970 to the average over 1951–1980, the years from 1971–1975 to the average over 1956–1985 and so on. The last 15 years (2001–2015) are compared to the average over 1986–2015 (see fig. 1c; compare calculation of ONI in Lindsey (2013)). We subtract these baseperiods from the original forcing for each pixel and identify anomalies associated with the type of El Niño according to table A1 (see fig. 1d). We used the ONI index to define the start, end and strength of the individual El Niño events and resampled the climate anomalies based on the ONI. We replaced anomalies in the climate forcing associated with El Niño events according to the best fit in duration and amplitude in ONI, i.e. events that start and end at a similar time in the year and have a similar timing and magnitude of the peak in ONI. For the replacement of the climate anomalies, we defined the start of an El Niño event as the second month of the first ONI season and the end as the second month of the last ONI season for each El Niño event (see fig. 1e and f). For example, the El Niño event from 1968–1969 started in the ONI-season September-October-November and ended in April-May-June, so the first month of the 1968/1969 El Niño event is the October in 1968 and the last month is March in 1969. Finally, we added the new anomalies and the original base periods to create the manipulated climate forcing.

This approach only isolates the effect of different expressions of El Niño to a limited extent since the calculated anomalies can also be influenced by other climate modes of variability. For example in Australia and Indonesia, different expressions of El Niño and different phases of the IOD can combine to drive the fire season (e.g. Pan et al., 2018). Given we create two synthetic forcings with respectively 15 CP (nine events replaced) and EP (eight events replaced) El Niño events, we assume that the emerging signal in the model results will be representative of the effect of different expressions of El Niño on the carbon balance.

We use the identical approach for the GSWP3 dataset. The only difference is the shorter length of the time period covered: GSWP3 ends in 2010, hence the last base period calculated is the average over 1981–2010 instead of 1985–2015 as for CRUNCEP.

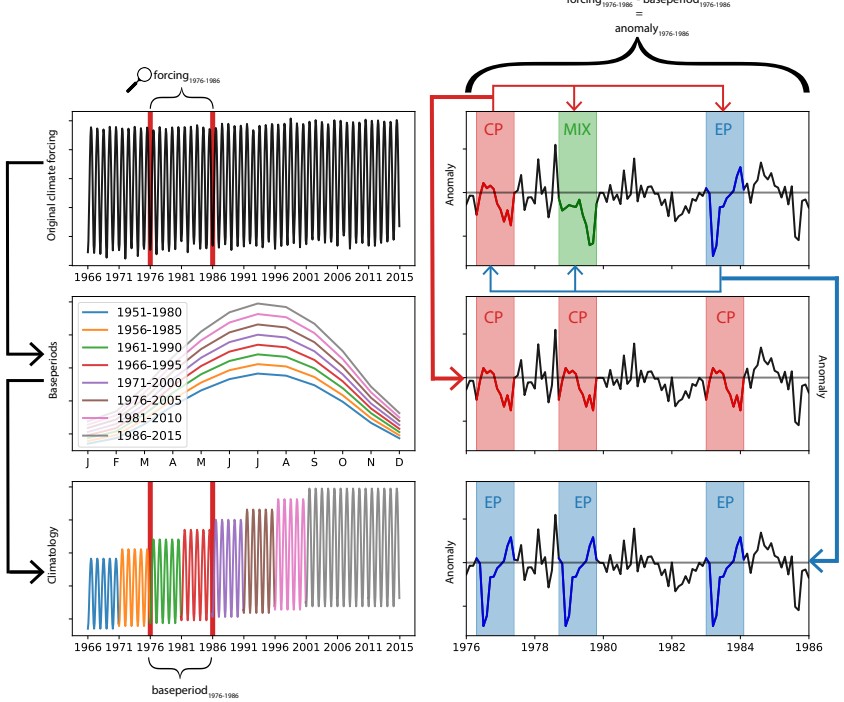

**Figure 1.** Schematic figure for the generation of the synthetic CP/ EP forcing datasets (see text for details).

We processed the data with netCDF Operators (NCO; version 4.7.7. http://nco.sf.net) and climate data operators (CDO; version 1.9.5. http://mpimet.mpg.de/cdo). The data analysis is conducted with python version 3.

## 3   Results

Figure 2 shows the effect of different expressions of El Niño on the net biome production (NBP). The upper three panels (fig. 2a, b, c) display total global NBP as well as tropical and Australian NBP for the control run and for the two experiments. All three runs have a similar magnitude in the IAV of NBP. Both the CP– and EP–only–scenarios can increase or dampen the peaks in NBP for the different regions. For all three regions, NBP accumulates to around 120 PgC globally, 80 PgC for the tropics and around 6 PgC for Australia (see fig. 2d–f). Overall, cycling of carbon (IAV of NBP) through the terrestrial biosphere was most marked in CP years relative to EP years (see fig. 2g, h and i). The magnitude in the difference of annual NBP in the CP– and EP–only-scenarios compared to the control run is comparable to the total IAV of NBP (compare fig. 2a, b, c). Overall, global changes in NBP accumulate to 9.6 PgC and 4.5 PgC for the CP– and the EP–only–scenario, respectively (see fig. 2g).

Figure 3 breaks down the NBP response to the terrestrial ecosystem fluxes gross primary production (GPP), terrestrial ecosystem respiration (TER; the sum of autotrophic and heterotrophic respiration) and fire emissions. Prior to 1997, individual CP events led to large IAV with increases in global GPP in some years of up to 7 PgC yr$^{-1}$ and reductions in some years of around $-0.5$ PgC yr$^{-1}$ (see fig. 3a). Changes in tropical GPP were mostly positive (up to 3.1 PgC yr$^{-1}$; see fig. 3b). By

contrast, in drier regions, for example in Australia, the year-to-year variability ranged between $-0.9$ PgC yr$^{-1}$ and 1.6 PgC yr$^{-1}$ (see fig. 3g). By comparison, TER varied by smaller amounts for all regions (see fig. 3b, e, h). Carbon fire emissions responded weakly to the expression of the El Niño (see fig. 3c, f, i). All fluxes show higher variability through to the end of the 20$^{\text{th}}$ century compared to the early 21$^{\text{st}}$ century. The lower variability in the 21$^{\text{st}}$ century coincides with a period of a positive phase in the interdecadal pacific oscillation.

However, the IAV does not lead to sustained trends in the ecosystem fluxes. The spatial distribution of the flux anomalies in the final year of the experiment (2013) displays spatial variability rather than systematic patterns, implying that the imposed changes also did not lead to long-term shifts in ecosystem processes at regional scales (see appendix figure B1).

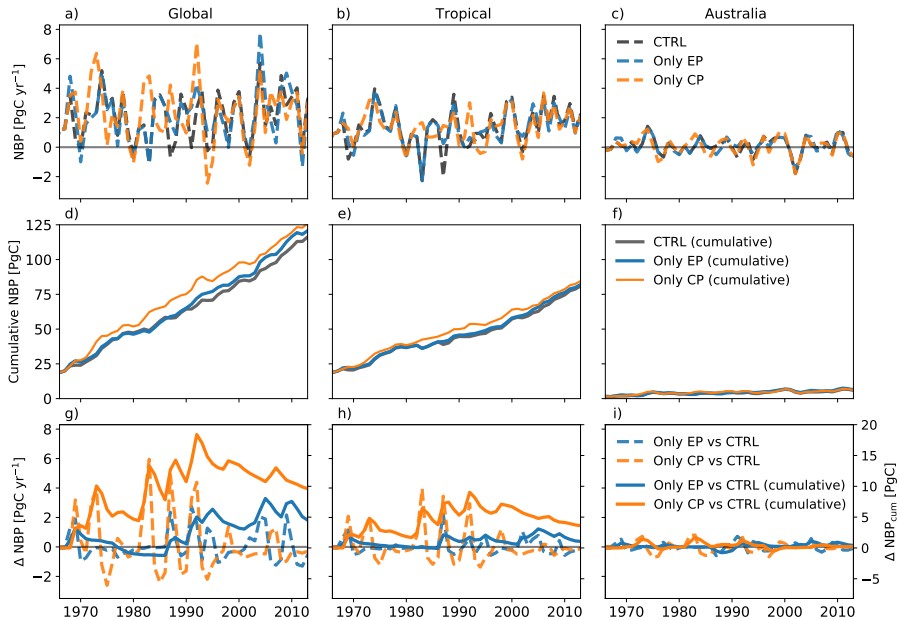

**Figure 2.** Total net biome production (NBP) (a-c), cumulative NBP (d-f) as well as absolute difference and cumulative sums of the difference between CP–only–scenario and control climate and EP–only–scenario and control climate (g-i).

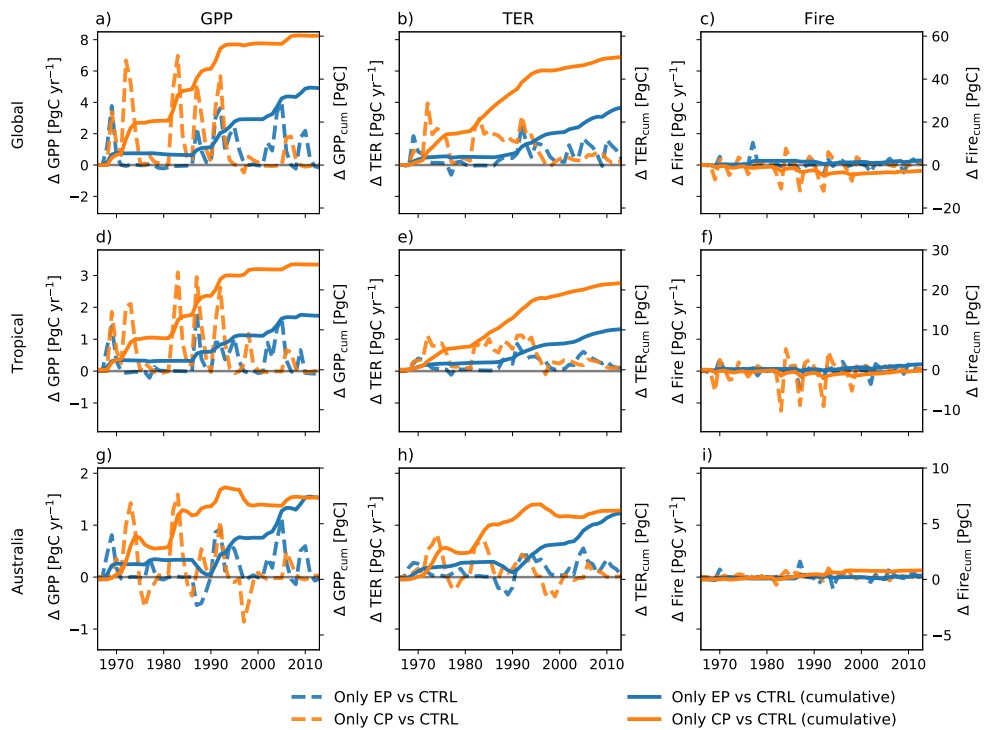

**Figure 3.** Absolute difference and cumulative sums of the difference between CP–only–scenario and control climate and EP–only–scenario and control climate for gross primary production (GPP), terrestrial ecosystem respiration (TER; the sum of autotrophic and heterotrophic respiration) and fire carbon emissions (Fire).

Figure 3 also shows the accumulated change in fluxes between 1968 and 2013. The cumulative sums of the absolute differences for fire carbon emissions are between −0.8 PgC and 0.3 PgC for the CP– and the EP–only–scenario, respectively (see fig. 3c). Over the 45 years, the accumulated GPP leads to a difference of 60.2 PgC and 35.8 PgC for the CP– and the EP–only–scenario, respectively, and this is largely balanced by the accumulated TER, 50.3 PgC and 26.6 PgC for the CP– and

170 the EP–only–scenario, respectively (see fig. 3a, b). For NBP, GPP and TER, a CP–only–scenario leads to stronger increases compared to an EP–only–scenario both globally and for tropical regions (see fig. 2g and h; see fig. 3a, b, d, e). In Australia, the cumulative sums of GPP and TER anomalies in an CP–only–scenario start to converge with the cumulative anomalies in a EP–only–scenario in 2005 for GPP and TER so that they reach similar values in 2013 (CP–only–scenario: 7.4 PgC for GPP and 6.2 PgC for TER; EP–only–scenario: 7.3 PgC for GPP and 5.9 PgC for TER; see fig. 3g, h). The cumulative carbon lost

175 through fires declines in an CP–only–scenario globally and in the tropical regions and is close to zero for Australia (see fig. 3c, f, i). In contrast to the absolute differences in the fluxes in the year 2013 (see above), cumulative GPP and TER show a clear(er) pattern with increases for both fluxes in southern South America and over Australia (see appendix figure B3). The accumulated increases in GPP however are balanced by increases in TER so that cumulative NBP shows strong spatial variability similar to the fluxes in appendix figure B1.

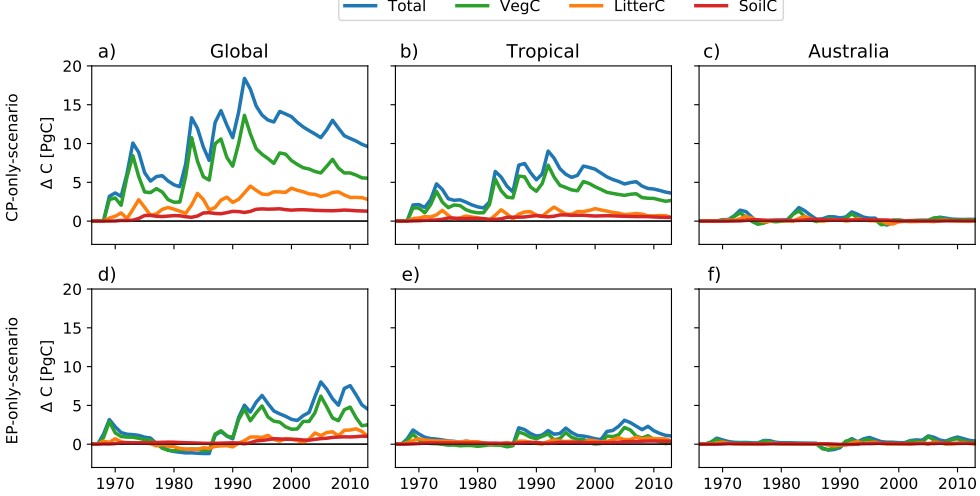

**Figure 4.** Absolute difference between CP–only–scenario ('CP–global') and control climate and EP–only–scenario and control climate ('EP–global') for the total, vegetation, litter and soil carbon pools.

At the global scale, the CP–only simulations led to an increase in the total land carbon storage (see fig. 4a). Between 1968 to 2013, the total carbon stored increased by 9.6 PgC compared to the control run (see fig. 4a; 'Total'; i.e. the sum of carbon stored vegetation, litter and soil). The EP–only simulations led to a gain of ∼4.5 PgC relative to the control run (see fig. 4d). Figure 4 also shows the breakdown of the change in carbon storage between the vegetation, litter and soil pools. In the CP–only simulation, the total change is dominated by an increase in vegetation carbon of ∼5.5 PgC originating from cumulative changes in GPP outbalancing those of TER. By contrast, in the EP–only simulations, any short-term increases in vegetation biomass are balanced by increased respiration, tissue turnover and mortality leading to a negligible change in ecosystem carbon storage. In both CP– and EP–only–scenario, the total differences in terrestrial carbon pools are largely the result of the responses of tropical ecosystems. Similar to the carbon fluxes, no clear patterns in the spatial distribution of the carbon pool anomalies emerge (see appendix figure B4).

## 4  Discussion

The El Niño Southern Oscillation (ENSO) strongly influences global and regional climate and has the potential to modify the regional and global carbon balance. Here, we examine whether two expressions of El Niño (CP and EP), as distinct from the El Niño phenomenon itself, modifies the regional and global carbon balance. This is timely: EP El Niño events might become more extreme in the future (e.g. Wang et al., 2019; Cai et al., 2018) and the occurrence of CP El Niño events seems to have increased over the later half of the 21st century and may increase further in the future (e.g. Yeh et al., 2009; Ashok et al., 2009). While the impact of more extreme (EP) El Niño events has been examined (e.g. Kim et al., 2017), there are few studies exploring the impact of different expressions of El Niño on the terrestrial carbon cycle. Previous work has focused on short timescales and explored either time lag effects on the carbon growth rate (Chylek et al., 2018), single regions and/ or single

events (Amazonia, Li et al. (2011); Indonesia, Pan et al. (2018)), or the composite anomalies in the carbon fluxes (Wang et al., 2018) in a larger spatial context. In effect, the response of ecosystems to different expressions of El Niño on longer timescales is not well understood.

In this study we show that, in line with previous studies (e.g. Wang et al., 2018; Chylek et al., 2018) climate anomalies associated with different expressions of El Niño have a strong impact on the IAV of ecosystem carbon fluxes. The El Niño-associated climate anomalies in our experiments do not show a consistent pattern but rather display high temporal variability between individual El Niño events (see appendix figure B5). Wang et al. (2018) showed that between El Niño events, the atmospheric $CO_2$ growth rate varied by 4 PgC yr$^{-1}$ at the peak for EP events and $\sim$2 PgC yr$^{-1}$ for CP events. Consequently, the ecosystem fluxes vary strongly in their response to different expressions of El Niño for individual years.

Despite the large resulting IAV in ecosystem carbon fluxes, the changes in GPP show a clear cumulative global increase of about 60 PgC (CP–only–scenario) and 36 PgC (EP–only–scenario) over the 45 years simulated (see fig. 3 a, d, g). In both scenarios, additional photosynthetic carbon uptake was mostly balanced by terrestrial ecosystem respiration (50 PgC for CP–only–scenario, 27 PgC for EP–only–scenario; see fig. 3 b, e, h). This, and the strong interannual variability, leads to small net changes in cumulative NBP over 45 years. The strong IAV in NBP therefore only results in a minor change in the total carbon storage simulated over 45 years, with 9.6 PgC more in an CP–only–scenario and $\sim$4.5 PgC more in the EP–only–scenario (see fig. 4 a, d).

Overall, the high spatial and temporal variability in the changes suggest that the effect of different expressions of El Niño on the terrestrial carbon cycle are important for predicting responses on interannual timescales (e.g. the atmospheric $CO_2$ growth rate) but are unlikely to affect the terrestrial carbon balance on longer timescales. Our model results imply that the anomaly patterns in the El Niño expression on climate forcing were too variable (and short-lived) to result in systematic shifts in vegetation composition. Nevertheless, the marked IAV of carbon fluxes implies an underlying sensitivity that may be particularly important for predictability of the carbon balance in drier ecosystems and/or water-limited agricultural regions. Interconnections between the terrestrial carbon cycle and ENSO have been widely explored (e.g. Zhang et al., 2019; Rödenbeck et al., 2018; Chylek et al., 2018). Whilst we also find key NBP variability on annual to decadal timescales (see fig. 2), particularly in CP years (accumulated NBP = 9.6 PgC), we did not find that this shorter timescale variability translated into sustained trends (1968–2013) in ecosystem fluxes, or shifts in vegetation distributions (see fig. 3).

## 4.1 Future directions

In our study we used a dynamic global vegetation model (DGVM) to examine the sensitivity of the terrestrial carbon cycle to changes in El Niño patterns. In response to climate, DGVMs predict global vegetation distributions based on plant physiology, competition, demography and vegetation structure (Sitch et al., 2003; Woodward and Lomas, 2004). In particular, these models also consider how fire dynamics and vegetation composition may respond to a shift in climate. In the past DGVMs have been widely used to examine how vegetation distributions may change in response to climate (Hickler et al., 2012; Martens et al., 2020) and fire (Kelley and Harrison, 2014). Since we only use a single model we cannot quantify uncertainties associated with alternative models and/or missing processes. For example, LPJ–GUESS, similar to many land surface and dynamic global

vegetation models, does not account for acclimation of plant respiration to increased temperature, and may consequently overestimate the carbon sensitivity to temperature changes on short timescales (e.g. Wang et al., 2020; Huntingford et al., 2017; Smith et al., 2015). Similarly, models differ in their sensitivity of the carbon cycle as water becomes limiting (Powell et al., 2013), which may affect the magnitude of carbon uptake in extreme El Niño years. Fisher et al. (2018) also highlighted hydrodynamics as well as the representation of demographic processes (e.g. recruitment and mortality) and fire disturbance as areas of uncertainty and promising for model development. Future experiments will also need to explore how rising $CO_2$ and temperature change the relative balance of GPP uptake and carbon losses via respiration during El Niño events. Wang et al. (2018) showed that the TRENDY model ensemble (which includes an LPJ family member) generally captured the NBP anomalies associated with CP El Niño events and only underestimates the anomalies associated with extreme EP El Niño events. This suggests results obtained with LPJ-GUESS would be broadly consistent with other DGVMs.

To place our results into a broader context, we examined whether LPJ–GUESS captures anomalies associated with different expressions of El Niño in the carbon cycle similarly to other models. We used the TRENDY v7 S2 run with transient $CO_2$ forcing and climate, but no imposed land use change. We choose the seven state of the art DGVMs CABLE-POP (Haverd et al., 2018), CLASS–CTEM (Melton and Arora, 2016), CLM5.0 (Oleson et al., 2013), JSBACH (Reick et al., 2013), LPX (Keller et al., 2017), OCN (Zaehle and Friend, 2010), ORCHIDEE (Krinner et al., 2005), ORCHIDEE–CNP (Goll et al., 2017), SURFEX (Boone et al., 2012) and VISIT (Kato et al., 2013) to calculate the TRENDY composite. LPJ–GUESS matches the TRENDY composite well for GPP and TER for the global, tropical and Australian averages with high correlation coefficients for the global and Australian averages (0.52–0.84) and low to moderate correlation coefficients for the tropics (0.17–0.6) except for the GPP anomaly associated with EP El Niño events (0.79) (see appendix figure B10). Similarly, the $R^2$–values are low for all tropical anomalies and global EP-anomalies (0–0.37), and low to moderate for the remaining regions (0.36–0.67). In general, LPJ–GUESS displayed greater variability than the TRENDY composite but is mostly within the model range (except for the GPP anomaly for EP El Niños; see appendix figure B10). The spatial distribution of the composite anomalies shows that LPJ–GUESS captures the features of anomalies in GPP associated with EP El Niño events compared to the individual models and the TRENDY model ensemble (see appendix figure B11). In contrast, LPJ–GUESS generally simulates weaker anomalies in GPP associated with CP El Niño events in Brazil and Western Africa compared to the ensemble mean and most individual models. This low sensitivity might also explain the relatively low correlation and $R^2$–values in appendix figure B10 for tropical regions and may dampen the overall response to the CP–only–scenario. We note however that LPJ–GUESS still is within the model range and can therefore be viewed as representative. In addition, LPJ-GUESS has a strong negative bias in Australia. As our results show, Australia does not make a large contribution to long-term changes in any of the carbon fluxes and pools. We also examined the sensitivity of our results to the use of a nitrogen cycle with LPJ-GUESS (see appendix figure B10), but did not find a strong sensitivity, most likely because nitrogen is not thought to be strongly limiting in the tropics (Vitousek, 1984). Based on this analysis, we suggest that our model sensitivity would likely be similar to that displayed by the other TRENDY models, although we would anticipate subtle regional differences, particular in the tropics if an alternative DGVM had been used. Especially for EP El Niño events, LPJ-GUESS diverges from the TRENDY ensemble mean that cannot be explained by nutrient limitation and suggests a different sensitivity to the meteorological drivers (see appendix figure B10).

Lastly, a comparison with satellite-derived observations might help to estimate whether LPJ-GUESS or indeed an alternative DGVM, captures the correct sensitivity in the response of vegetation dynamics to ENSO events. Nevertheless, as direct global measurements of carbon fluxes do not exist, and those that do are often based on models themselves, future work might restrict comparison to less direct proxies of variability e.g. leaf area index (Zhu et al., 2013) and/or GRACE terrestrial water storage (Rodell et al., 2004).

A further research path may consider driving a model with a larger ensemble of meteorological forcing to account for uncertainties associated with global climate reanalysis products. We conducted the same experiment based on the GSWP3 climate forcing and found that the overall variability in all terrestrial ecosystem flux and carbon pool anomalies is similar compared to the experiment based on the CRUNCEP dataset but with a smaller magnitude (see appendix figures B8 and B9). Wu et al. (2017) showed that the simulated GPP by LPJ-GUESS could vary by as much as 11 PgC yr$^{-1}$ globally, due to the use of alternative climate forcing data sets. Nevertheless, in their analysis Wu et al. (2017) showed that overall, the magnitude of tropical GPP was largely robust to the use of different precipitation forcing, although there was variation regionally. Moreover, exploring the impact of different expressions of El Niño in a future climate would be worthwhile. However, we note that this would probably require multiple DGVMs to account for the uncertainty associated with the vegetation responses to $CO_2$ and interactions with nutrients (Zaehle et al., 2014). In addition, the representation of ENSO diversity in CMIP5 and CMIP6 models is highly uncertain due to model biases, especially in the equatorial Pacific, resulting in low model agreement (e.g. Freund et al., 2020). Therefore, to obtain robust results, a future experimental design would also require an ensemble of climate forcing input datasets.

In this study, we run LPJ–GUESS with active stochastic and fire disturbance. Including these two types of disturbance contributes significantly to the spatial variability (compare appendix figures B1 and B2). Our results show that the fire patterns in LPJ–GUESS are largely insensitive to the imposed changes due to the expression of El Niño, which is in contrast with observational studies that suggest that El Niño events themselves are strongly linked to fire activity on regional scales (e.g. Pan et al., 2018; Mariani et al., 2016; Harris and Lucas, 2019; Fonseca et al., 2017). This might result from changes imposed in the experiment being too small to trigger changes in fire patterns. We note however that the fire module implemented in LPJ–GUESS (Thonicke et al., 2001, LPJ–GUESS–GlobFIRM; see) is a relatively simple empirical model that does not capture observed fire properties well (Hantson et al., 2020) and might underestimate the sensitivity of fire occurrence to different expressions of El Niño. Teckentrup et al. (2019) highlighted notable differences among seven DGVMs in the pattern of burned area to climate forcing. Our results suggest that the interaction between the expression of El Niño and fire requires further investigation.

Finally, isolating the effect of El Niño on the atmosphere and terrestrial biosphere is not trivial for individual events. Individual El Niño events vary in location, timing and magnitude (e.g. Capotondi et al., 2015) and teleconnections are influenced by the background climate and climate variability (e.g. the Indian Ocean Dipole). In our study, we assume that replacing a CP event with an EP event, or vice versa, did not modify the role played by other modes of variability. We further neglect possible interactions between consecutive ENSO events. For example, strong El Niño events tend to peak in the eastern Pacific, and these tend to be followed by a La Niña event. However, the influence of a preceding El Niño on the characteristics of the La

Niña event is not clear (Santoso et al., 2017). By generating two experiments with either 15 (nine manipulated events) CP or 15 (eight manipulated events) EP El Niño events, we assume that the signal observed at the end of the time period is driven by the respective expression of El Niño. In order to test the validity of our results, we applied a different approach where we replaced the climate forcing of CP El Niño years with the climate forcing of the EP El Niño events closest in time and found even smaller changes in carbon fluxes (see appendix figure B6) and in carbon sequestration (see appendix figure B7). An alternative approach could be to calculate composite anomalies for both CP and EP El Niño events and use those for replacement, but this would dampen variability in the forcing and introduce a different bias. Alternatively, generating a sea surface temperature forcing representing the different expressions of El Niño and using an atmospheric model to generate the climate anomalies that result from the changes in sea surface temperatures could help quantify the effect of the expression of El Niño on the carbon sequestration. However, given the changes we found are very small and spatially variable we doubt this would lead to different conclusions.

## 5 Conclusions

We explored the impact of the expression of El Niño on the terrestrial carbon cycle on multi-decadal timescales using LPJ–GUESS. We found that the changes in simulated anomalies reflecting the two expressions of El Niño in NBP accumulate around 9.6 PgC (CP–only–scenario) and 4.5 PgC (EP–only–scenario) respectively. However, this accumulation period covers more than 45 years and is therefore negligible compared to annual anthropogenic emissions of 9.4 +-0.5 PgC yr$^{-1}$ (Le Quéré et al., 2018). Our results therefore suggest that the impact of different expressions of El Niño on the carbon cycle on long timescales is likely to be small.

Our results imply that simulations of the terrestrial carbon cycle over the recent past and into the future using global climate models may not require the expression of El Niño events to be well captured. There are major challenges in accurately capturing El Niño - La Niña cycles and the teleconnections associated with El Niño events with existing climate models. Had we found the expression of El Niño to be critical in simulating the long term terrestrial carbon balance, this would have added a very significant additional uncertainty to projections of the future role of the land in storing carbon. Our results suggest that the expression of El Niño, as distinct from whether there is an El Niño or a La Niña, is relatively unimportant over the long term. We note that our results do agree with earlier studies (Chylek et al., 2018; Wang et al., 2018; Pan et al., 2018) that the expression of El Niño is important to terrestrial carbon fluxes on shorter, annual and interannual timescales. Overall, in the context of the long-term global and regional terrestrial carbon balance, our results imply that model development should prioritise simulating El Niño - La Niña cycles and the associated teleconnections, with perhaps less focus needed on considering the additional challenge of resolving the expression of individual El Niño events.

*Code and data availability.* The analysis codes are available at https://github.com/lteckentrup/nino_experiment). The model code is available upon request from http://web.nateko.lu.se/lpj-guess/contact.html. The model outputs will be shared in line with UNSW's open-access policy

**Table A1.** El Niño events from 1968–2010 identified by the NOAA Oceanic Niño Index (ONI) and their different expressions derived by four methods according to Yu and Kim (2013): pattern correlation method ('PTN'; Yu and Kim, 2013); central location method ('Niño'; Kug et al., 2009; Yeh et al., 2009), the El Niño Modoki Index ('EMI'; Ashok et al., 2007) and the cold tongue/ warm pool index ('CT/WP'; Ren and Jin, 2011). We define a CP or EP El Niño where three out of the four indices agree on the same El Niño type. The remaining events are defined as mixed events ('MIX').

| Year | Dominant El Niño type | CP El Niño replacement | EP El Niño replacement |
|---|---|---|---|
| 1968–1969 | CP | – | 1976–1977 |
| 1969–1970 | EP | 1977–1978 | – |
| 1972–1973 | EP | 2009–2010 | – |
| 1976–1977 | EP | 1977–1978 | – |
| 1977–1978 | CP | – | 1976–1977 |
| 1982–1983 | EP | 2009–2010 | – |
| 1986–1987 | EP | 1994–1995 | – |
| 1987–1988 | MIX | 2002–2003 | 1982–1983 |
| 1991–1992 | MIX | 2009–2010 | 1997–1998 |
| 1994–1995 | CP | – | 1976–1977 |
| 1997–1998 | EP | 2002–2003 | – |
| 2002–2003 | CP | – | 2006–2007 |
| 2004–2005 | CP | – | 1976–1977 |
| 2006–2007 | EP | 1977–1978 | – |
| 2009–2010 | CP | – | 1972–1973 |

on publication. The TRENDY version 7 model output is available upon request (https://sites.exeter.ac.uk/trendy) and the CRUNCEP climate
forcing is available from https://rda.ucar.edu/datasets/ds314.3/.

# Appendix B: Figures

## B1 CRUNCEP

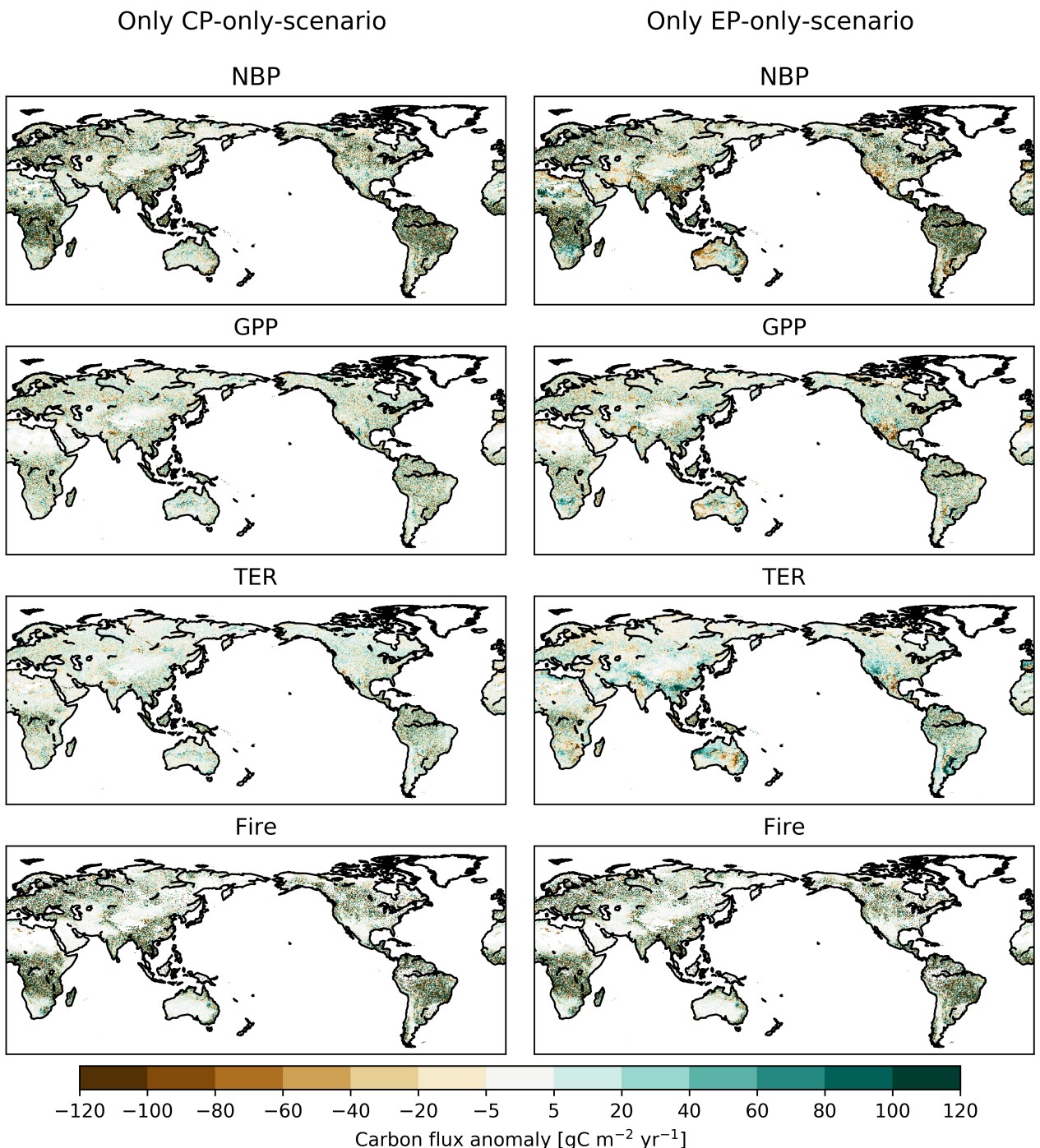

**Figure B1.** Absolute difference between CP–only–scenario and control climate and EP–only–scenario and control climate for net biome production (NBP), gross primary production (GPP), terrestrial ecosystem respiration (TER) and fire carbon emissions (Fire) for the final year of the experiment (2013). Note that the noise partially results from stochastic and fire disturbance (compare fig. B2).

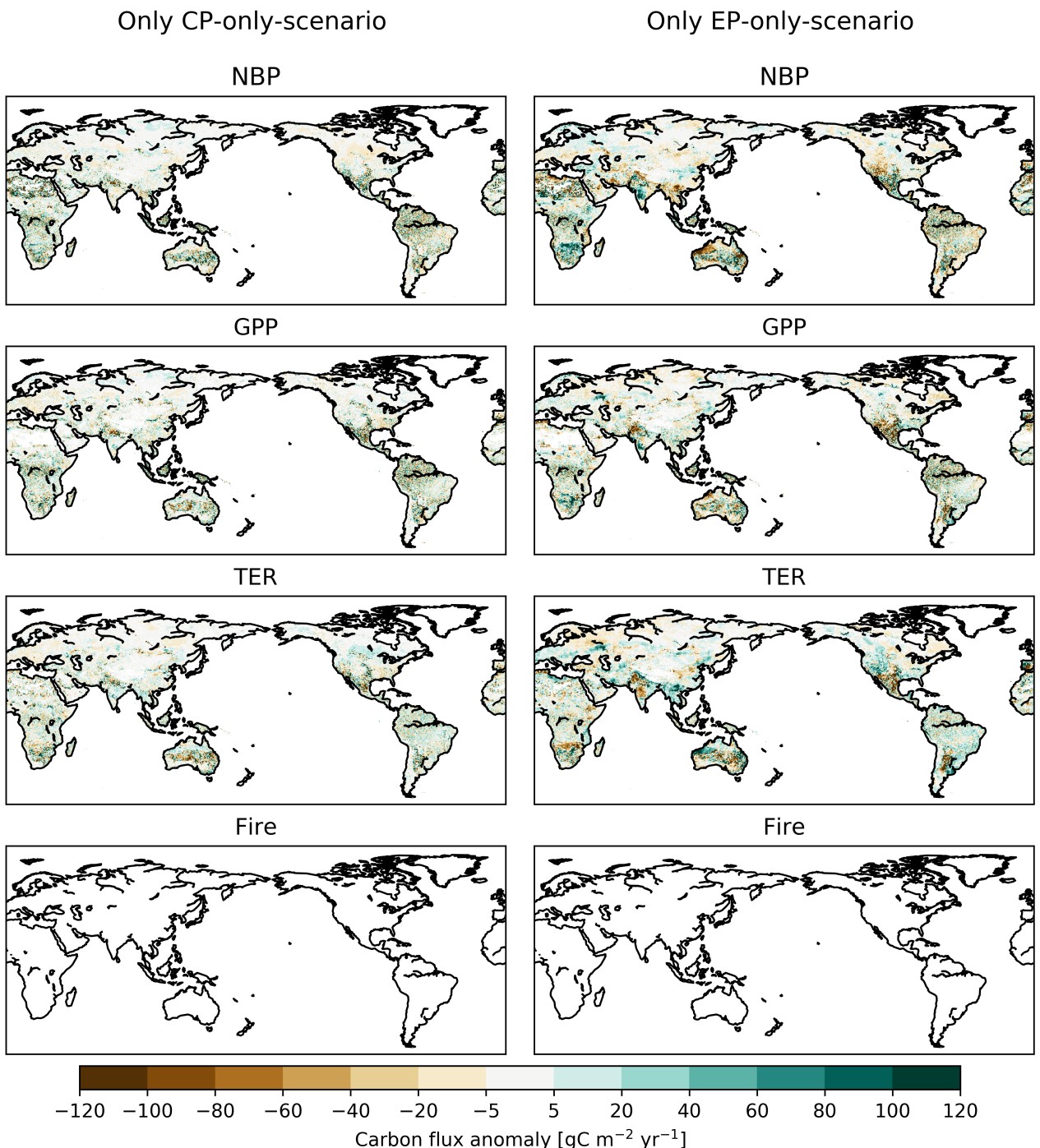

**Figure B2.** Absolute difference between CP–only–scenario and control climate and EP–only–scenario and control climate for net biome production (NBP), gross primary production (GPP), terrestrial ecosystem respiration (TER) and fire carbon emissions (Fire) for the final year of the experiment (2013) without stochastic and fire disturbance.

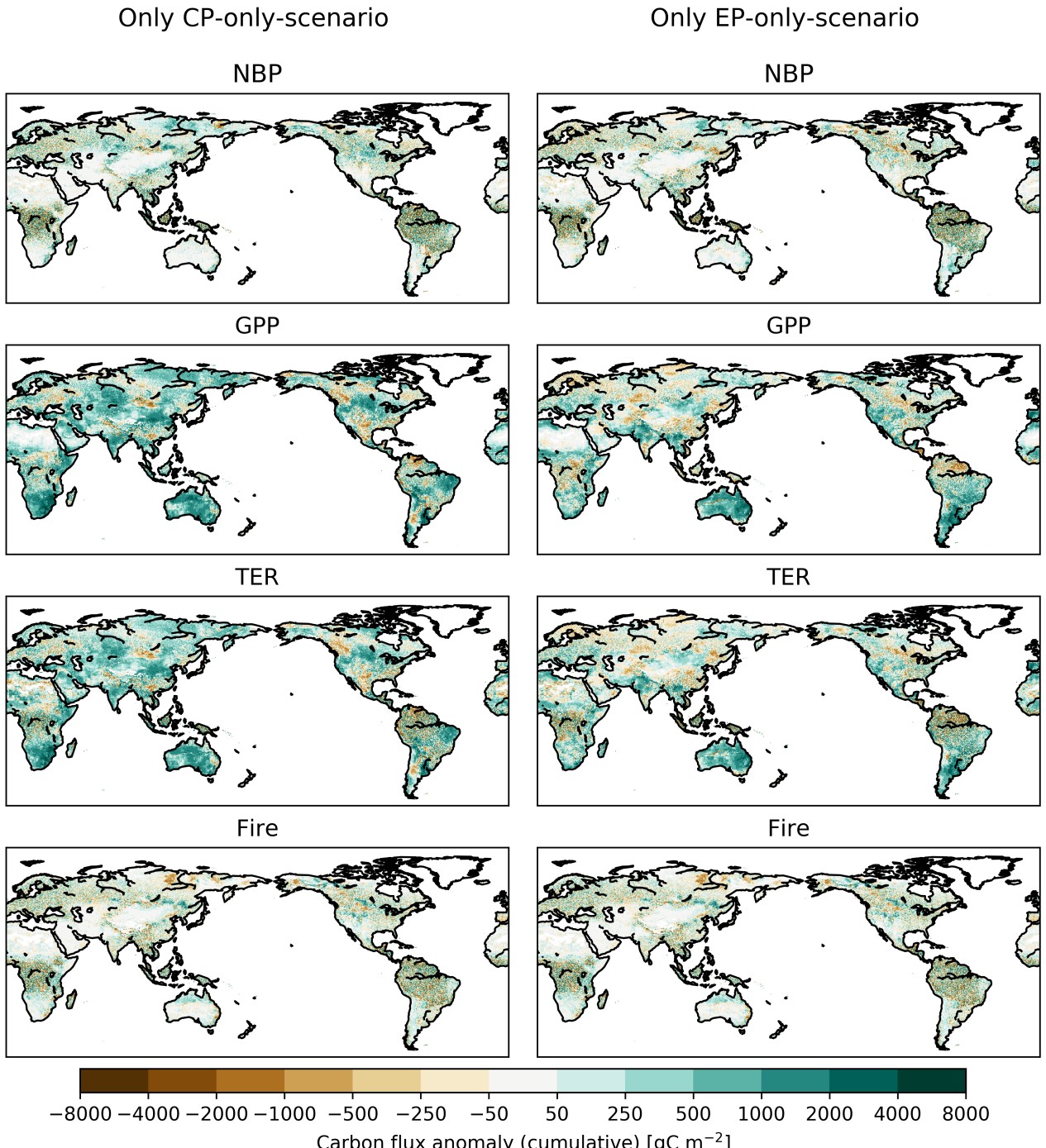

**Figure B3.** Accumulated absolute difference between CP–only–scenario and control climate and EP–only–scenario and control climate for net biome production (NBP), gross primary production (GPP), terrestrial ecosystem respiration (TER) and fire carbon emissions (Fire) for the final year of the experiment (2013).

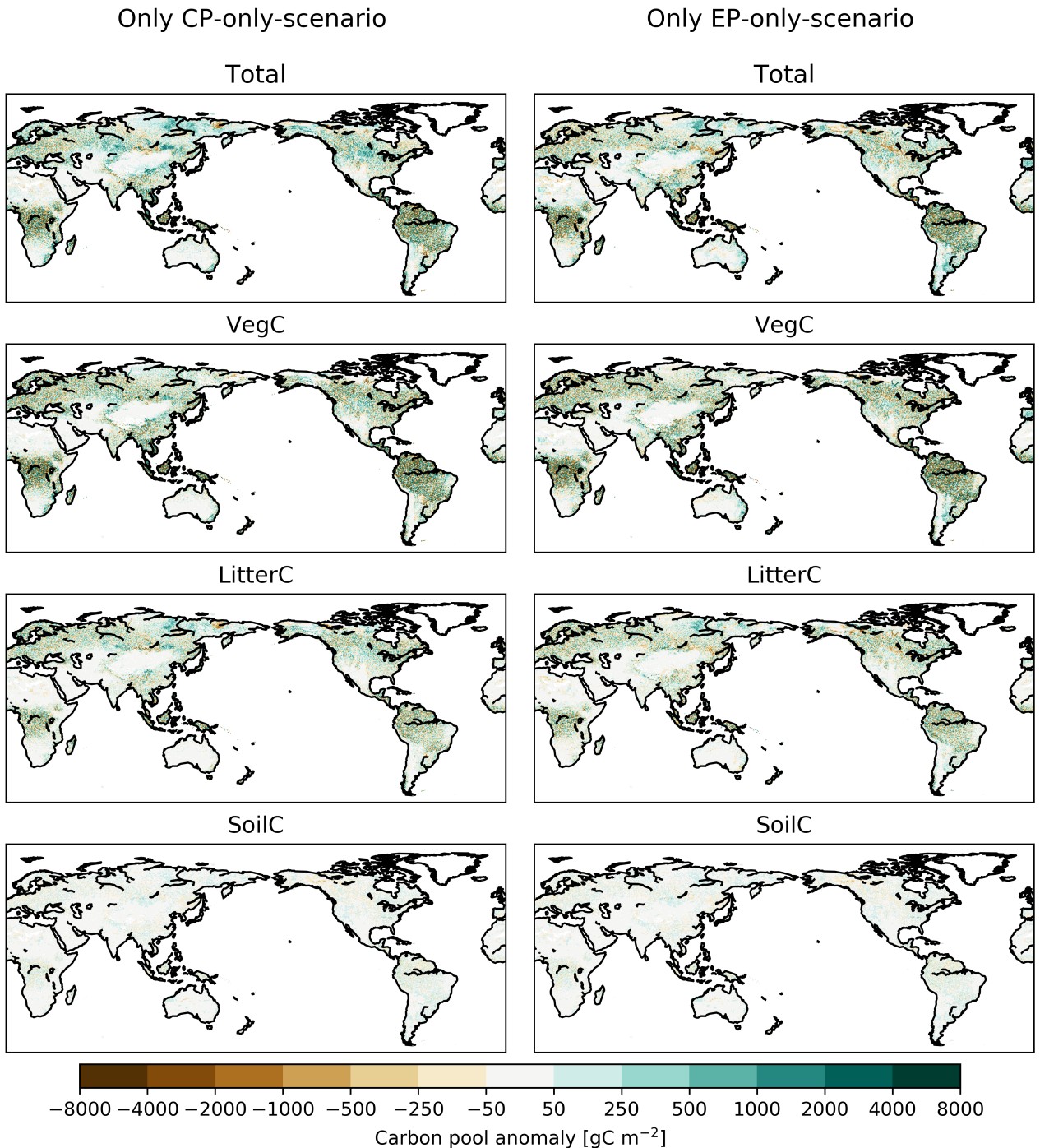

**Figure B4.** Absolute difference between CP–only–scenario and control climate and EP–only–scenario and control climate for total, vegetation, litter and soil carbon pool for the final year of the experiment (2013).

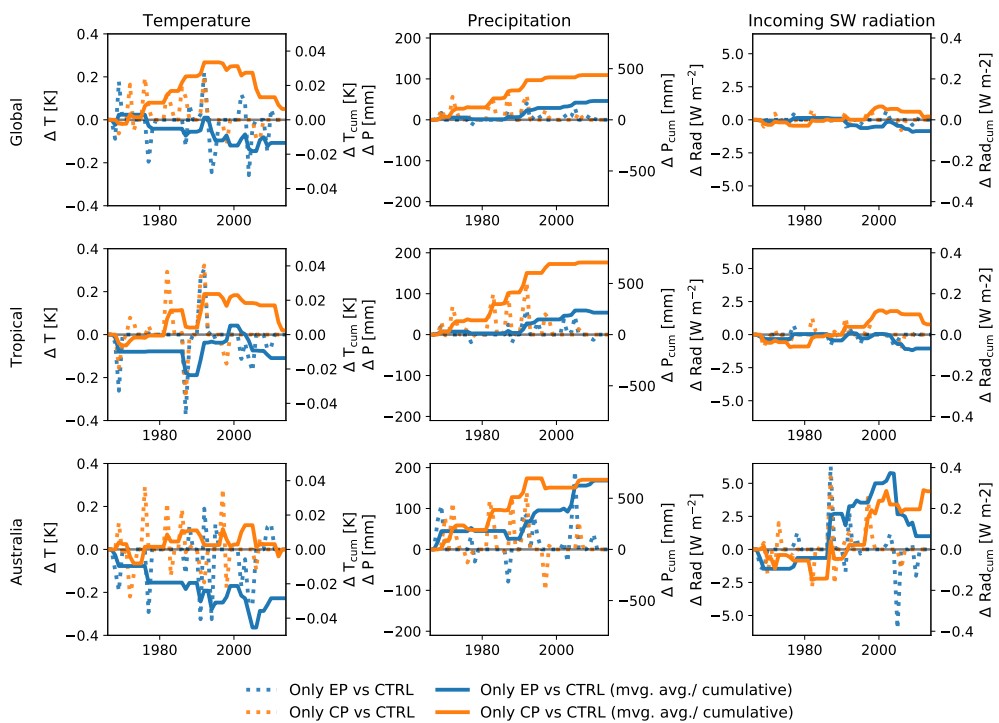

**Figure B5.** Absolute difference and cumulative sums of the difference between CP–only–scenario and control climate and EP–only–scenario and control climate for precipitation and absolute difference and 30 year moving average of the difference between CP–only–scenario and control climate and EP–only–scenario and control climate for temperature and incoming shortwave radiation.

## B2 Nearest year replacement

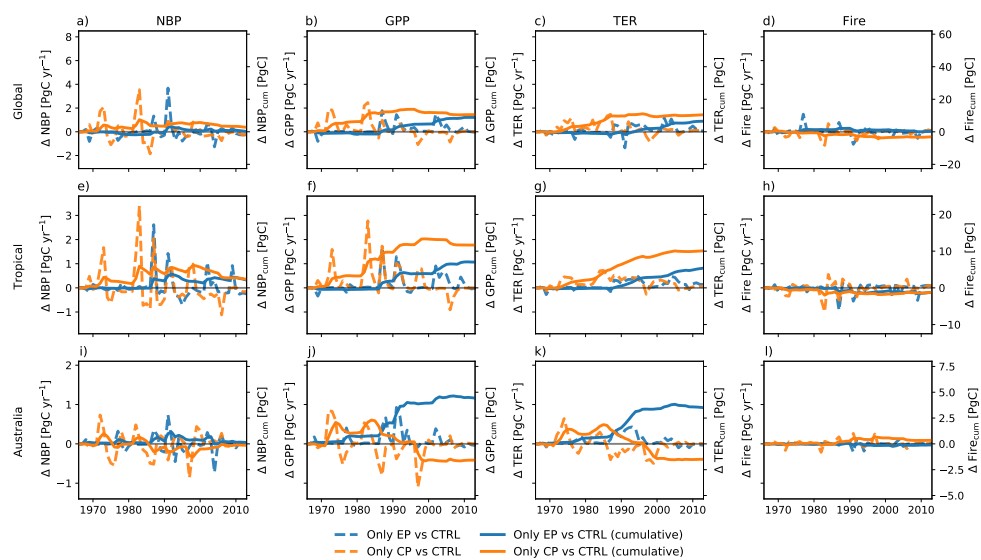

**Figure B6.** Absolute difference and cumulative sums of the difference between CP–only–scenario and control climate and EP–only–scenario and control climate for net biome production (NBP), gross primary production (GPP), terrestrial ecosystem respiration (TER; the sum of autotrophic and heterotrophic respiration) and fire carbon emissions (Fire) for an alternative method (replacing the climate forcing of CP El Niño event with that of an EP El Niño event and vice versa for events closest in time).

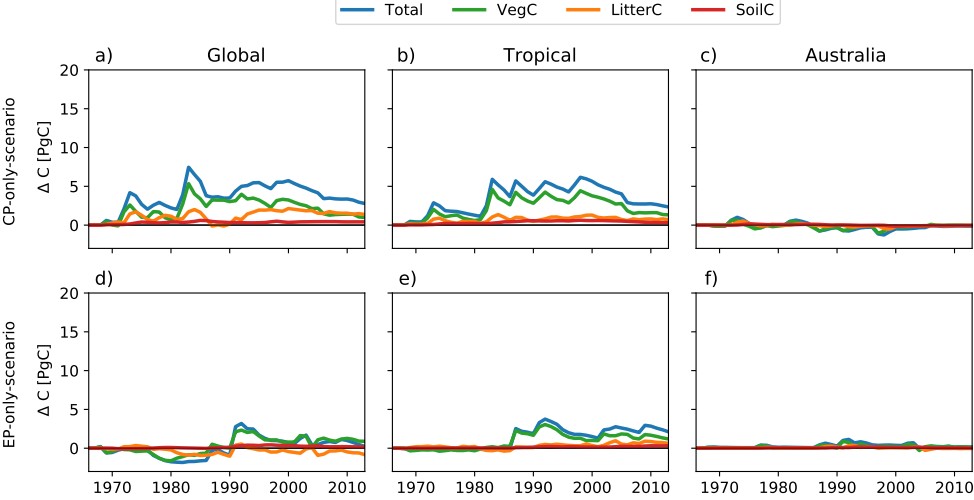

**Figure B7.** Absolute difference between CP–only–scenario ('CP–global') and control climate and EP–only–scenario and control climate ('EP–global') for the total, vegetation, litter and soil carbon pools for an alternative method (replacing the climate forcing of CP El Niño event with that of an EP El Niño event and vice versa for events closest in time).

## B3    GSWP3 forcing

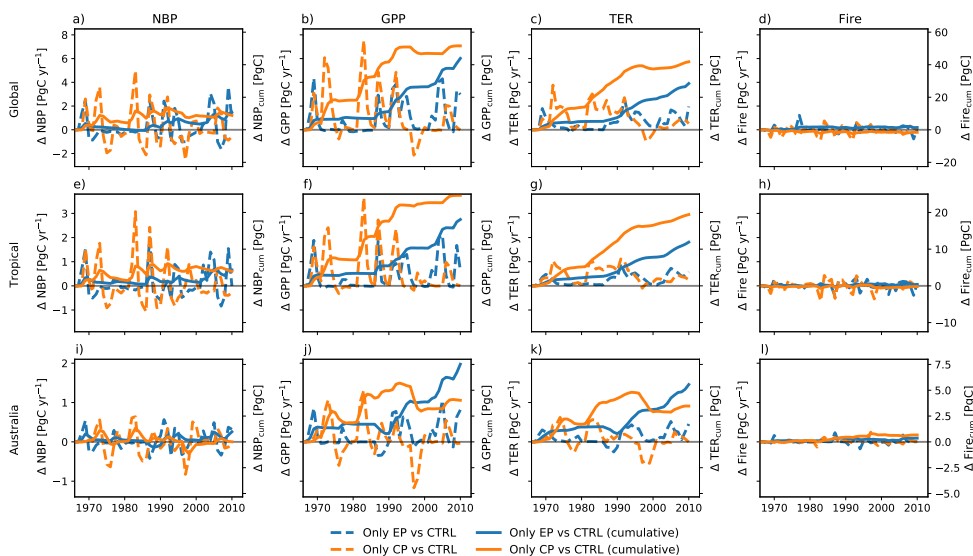

**Figure B8.** Absolute difference and cumulative sums of the difference between CP–only–scenario and control climate and EP–only–scenario and control climate for net biome production (NBP), gross primary production (GPP), terrestrial ecosystem respiration (TER; the sum of autotrophic and heterotrophic respiration) and fire carbon emissions (Fire) for the experiments based on the GSWP3 forcing.

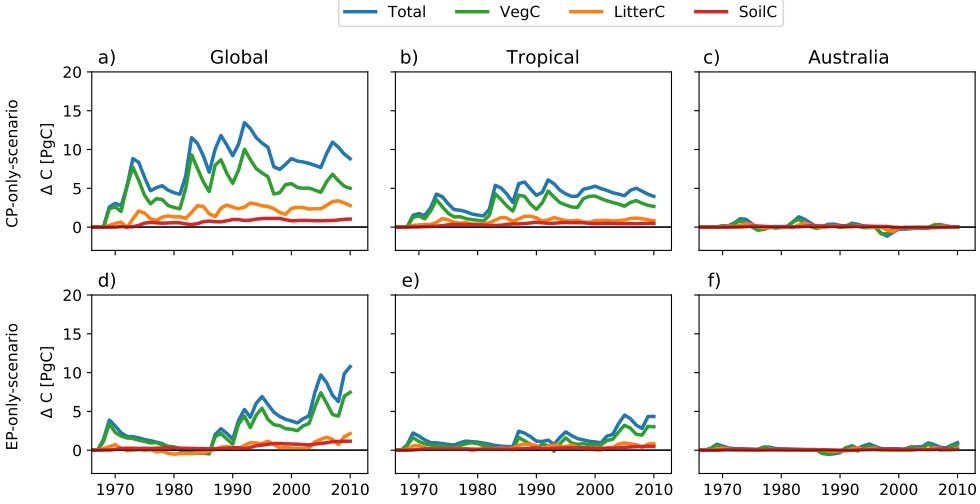

**Figure B9.** Absolute difference between CP–only–scenario ('CP–global') and control climate and EP–only–scenario and control climate ('EP–global') for the total, vegetation, litter and soil carbon pools for the experiments based on the GSWP3 forcing.

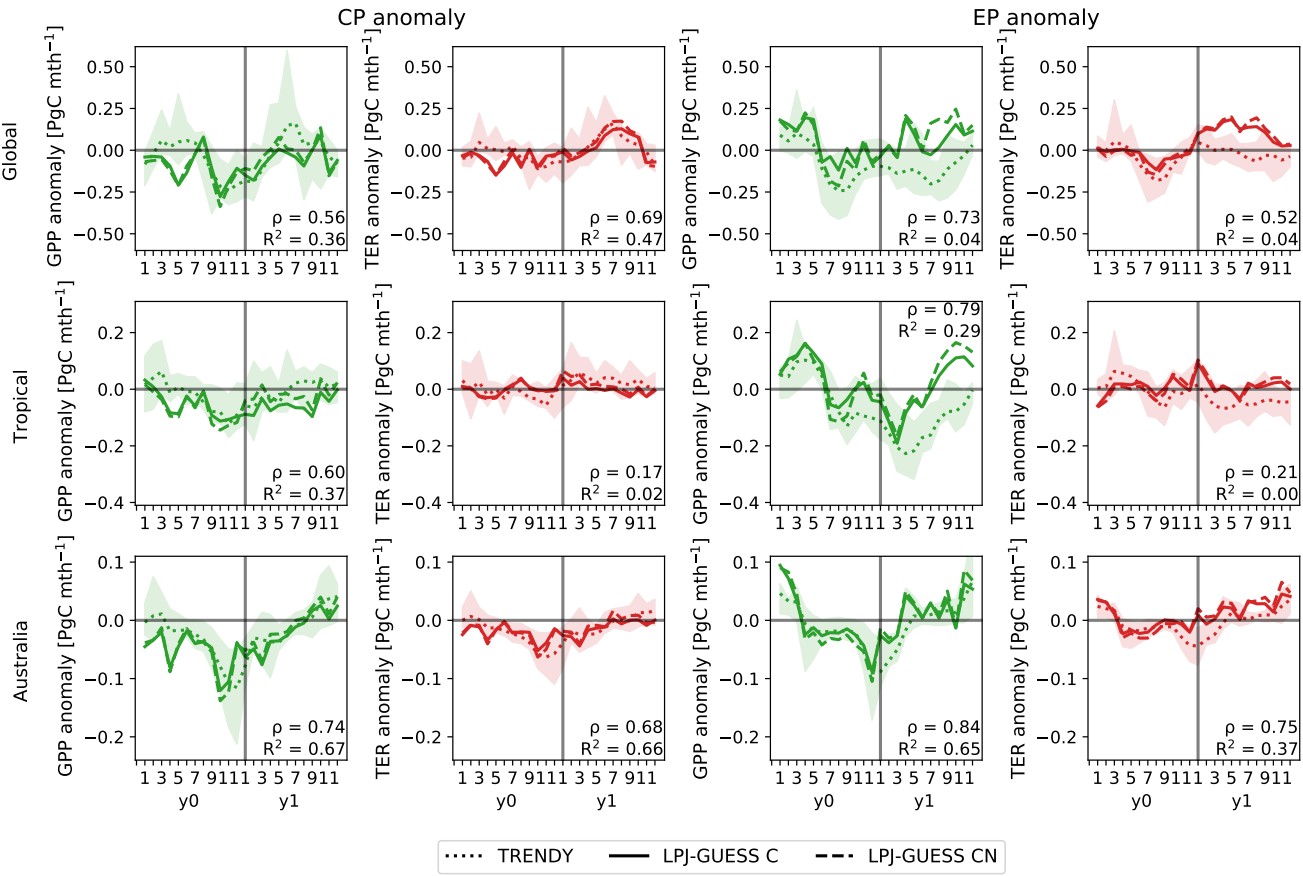

**Figure B10.** Monthly composite anomalies during the El Niño developing (y0) and decaying (y1) year in gross primary production (GPP; green lines) and terrestrial ecosystem respiration (TER; sum of autotrophic and heterotrophic respiration; red lines) for all CP and EP El Niño events listed in appendix table A1 averaged over the globe, the tropics (23°S –23°N) and Australia. The dotted lines show the TRENDY V7 composite, the solid lines are the individual LPJ–GUESS runs without nitrogen cycling ('LPJ-GUESS C') and the dashed lines show the individual LPJ–GUESS runs with nitrogen cycling ('LPJ-GUESS CN') (compare Wang et al., 2018). The shaded area shows the model spread of the individual TRENDY models. $\rho$ and $R^2$ are the Pearson correlation coefficient and the $R^2$ value respectively between the individual LPJ-GUESS runs without nitrogen cycling and the TRENDY ensemble mean.

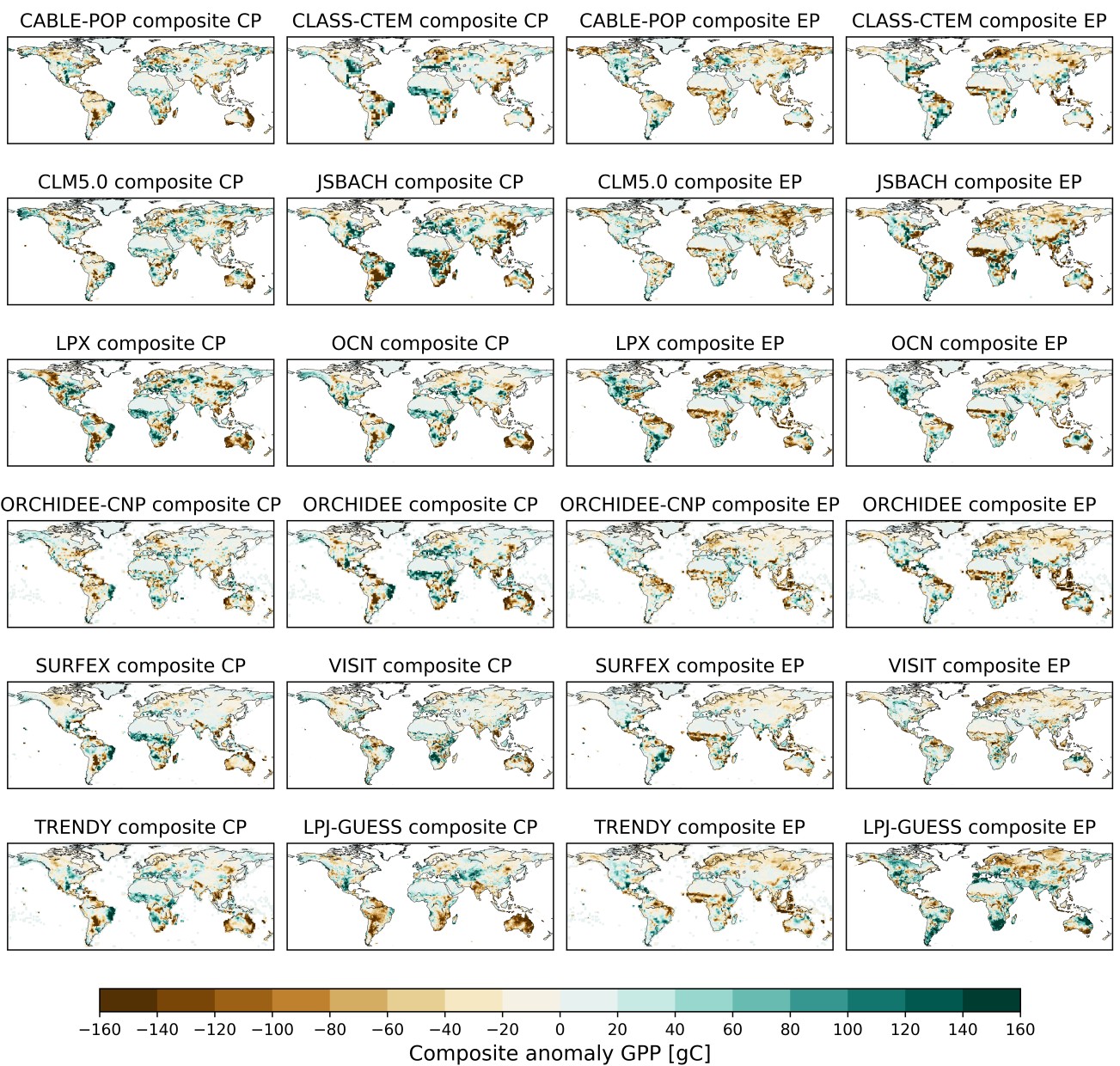

**Figure B11.** Composite anomalies in gross primary production (GPP) summed over the the El Niño developing and decaying year for all CP and EP El Niño events listed in tab. B1 for the individual TRENDY models, the TRENDY composite and the individual LPJ–GUESS run (compare Wang et al., 2018).

*Author contributions.* LT, MDK and AJP designed the experiment and performed the model runs and the analysis with input from BS. LT, MDK and AJP wrote the paper with contributions from BS.

*Competing interests.* The authors declare that they have no conflict of interest.

*Acknowledgements.* LT, MDK and AJP acknowledge support from the Australian Research Council (ARC) Centre of Excellence for Climate Extremes (CE170100023). MDK and AJP acknowledge support from the ARC Discovery Grant (DP190101823). MDK was also supported from the NSW Research Attraction and Acceleration Program. We further acknowledge the TRENDY DGVM community, as part of the Global Carbon Project, for access to their model outputs. We thank the National Computational Infrastructure at the Australian National University, an initiative of the Australian Government, for access to supercomputer resources. Finally, we thank Matthew Forrest for his support in running LPJ–GUESS.

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
