# Peer review of "Examining the sensitivity of the terrestrial carbon cycle to the expression of El Niño"

_Biogeosciences, 2020_

## Short Comment (SC1) · 20 Aug 2020

The forcing of El Nino is a combination of the annual cycle and the tidal cycles. There are peer-reviewed citations for this but an intriguing terrestrial connection is to predator-prey cycles, see figure below and the following cite: Archibald, H. L. "Relating the 4-year lemming ( Lemmus spp. and Dicrostonyx spp.) population cycle to a 3.8-year lunar cycle and ENSO". Can. J. Zool. 97, 1054–1063 (2019).
* * *
[Figure]

[Figure]

**Fig. 1.**

---

## Short Comment (SC2) · 28 Aug 2020

This is a very interesting study. A related study on the impact of central pacific El Nino and canonical El Nino on carbon fluxes are published in Valsala, V., M. Roxy, K. Ashok, and R. Murtugudde, Spatio-temporal characteristics of seasonal to multidecadal variability of pCO2 and air-sea CO2 fluxes in the equatorial Pacific Ocean,2014, Journal of Geophysical Research, 119, DOI:10.1002/2014JC010212, 8987-9012 may be kindly cited. Both these studies share a common scientific subject (impacts of CP and EP El Nino in the carbon cycle), therefore it is appropriate to mention above study in this paper as well.

Thanks, -Vinu

---

## Referee Comment (RC1) · Anonymous Referee #1 · 20 Sep 2020

This manuscript investigates the impacts of different expressions of El Nino on the long-term terrestrial carbon storages, using a DGVM LPJ-GUESS with the manipulated climate forcing. They pointed out that CP and EP events can significantly influence the interannual variability of terrestrial carbon cycle, but cannot lead to NBP trend. Therefore, they suggest that future simulations of carbon cycle may not need to well simulate the expressions of El Ninos in Earth System model. The method is well described and writing is clear with concise and clear conclusions. Some minors: (1) L120: "associated with El Nino events according to the best fit in duration and amplitude in ONI . . .". Because there are actually 6 CP, 7 EP, and 2 Mix, you can clearly show the replacement relationships in the table for the manipulations (like in Table A1). It can be more straightforward for us to understand it. (2) The units in spatial patterns in Figure B1–B4

are not correct. For example, flux is gC=>gC/m2/yr?, carbon pool is gC yr-1?=>gC? (3) In Discussion: Some aspects can be mentioned further. a) ENSO diversity (Capotondi et al., 2015): Although replace the CP and EP events based on their durations and amplitudes, every ENSO event is unique with different spatial impacts. b) Changes in frequency of ENSO occurrence in future: Though it maybe doesn't influence your conclusions, you can discuss that frequency change may have some influences.
* * *

---

## Referee Comment (RC2) · Anonymous Referee #2 · 28 Sep 2020

In this manuscript, Teckentrup et al. used LPJ-GUESS forced by manipulated climate datasets to study the influences of two expressions of El Nino (CP and EP) on the terrestrial carbon cycle. The authors suggested that the expressions of El Nino only influence interannual variability of NBP (e.g. CP caused larger IAV in NBP than EP at the global scale) but not the long-term change in NBP. They concluded that the relative frequency of CP and EP is not critical in models as CP/EP did not yield detectable changes in long-term NBP. The science question is interesting, the story is well told and there is no major flaw in the method. That being said, there are a few questions that puzzled me after reading the manuscript, which I hope the authors could clarify a bit before I could support it.

1. One of the novel points presented is that "impact (of CP and EP) on longer

timescales is not well understood". El Nino, either CP or EP, is known to dominate the interannual variability of terrestrial carbon cycling. It is not clearly stated in the Introduction why we would expect an influence of CP/EP El Nino at longer timescales in the first place. In another word, would it be a surprise that CP/EP El Nino exert no change on long-term NBP, as we already known that El Nino influences IAV rather than long-term variability of the carbon cycle. Perhaps the relative more frequent CP occurrences in the future could be an issue long term but the current models may not include proper mechanisms (i.e. shift in species composition, acclimations) to answer the question.

2. The study is aimed at studying the sensitivity of the terrestrial carbon cycle to CP/EP El Nino. And the author did so by replacing the climate anomalies during CP to EP and vice versa. CP is reported to cause larger global IAV than EP. My concerns is: (using global simulation as an example) is this larger sensitivity of the terrestrial carbon cycle to CP is due to the changes in the inherent climate sensitivity of carbon during CP/EP, or is this simply caused by the generally larger climate anomalies during CP (Fig. B5). I would assume the reason is the latter, as the inherent climate sensitivity of carbon cycle is essentially predefined by the model (in this case LPJ-GUESS) structure, so what we see here (IAV of NBP in CP > EP) is perhaps just because the IAV of climate in CP > EP.

3. missed chance on the spatial and phenology of carbon fluxes. While I have doubts about the reported difference between CP and EP at interannual or longer time scales, I feel their difference is perhaps more pronounced at seasonal scales and spatial, when CP and EP show apparent contrasting temporal patterns (e.g. Fig 1). As was also noted by Clylek et al. 2018, the time delay of CO2 rise after SST increase is one of the pronounced differences, and the difference is only around 3 months. Focusing on longer time scales might easily just averaged out these important characteristics. I think the authors have done a nice job in demonstrating the spatial difference of carbon sinks under CP/EP, and these results perhaps worth more highlights. With that, I would

also say it maybe a stretch to say CP/EP is not critical in future models, as their major difference is likely to be clearer seasonally and spatially (e.g. different carbon sink distribution, phenology of carbon uptake).

Some minor issues:

L11. Please specify what kind of longer time scale effect (i.e. decadal mean, decadal variation or trend?) L84 and L104. If CRU-NCEP v7 covers 1901-2016, why not consider the 2015/2016 El Nino in the analysis. L84. By saying CRU, did you mean CRU-NCEP. L119-120. I am not sure I understand how to choose the replacements for CP and EP correctly. Why there is a need to resample climate anomalies using ONI and how do we locate the CP that is used to replace a EP (in the same 10-year window shown in Fig 1?). L210. Does LPJ-GUESS have a component to simulate species composition? B1-B4: Unit of carbon fluxes in supplementary figures. Per m2?

---

## Referee Comment (RC3) · Anonymous Referee #3 · 19 Oct 2020

Thank you for inviting me to review paper "Examining the sensitivity of the terrestrial carbon cycle to the expression of ElNino" by Teckentrup et al.

First, may I apologise for taking longer than the expected four weeks to return the review. I realise it can be unfair on the authors to have the Comments section closed, and then another further review appears. For that reason, I have tried to make the review a "light touch", and predominantly suggestions for better framing of the analysis in the future work part.

Possibly the most refreshing feature of this paper is that it actually has the confidence to present a "negative result". That is, for the processes investigated by factorial methods, these are likely to have a size that is relatively small compared to the overall impacts of on-going background climate change caused by fossil fuel burning. That is, though,

still really important to know, and it does not diminish from the paper.

However, by presenting the findings as unimportant also feels like a disservice to the paper findings? As so much recent research into the climate system illustrates, the simultaneous interannual variability of Earth System components does reveal much about potential long-term changes under global warming. Indeed the entire Emergent Constraint concept is based on such an approach. Hence, when placed in that context, the quite specific findings of this analysis become particularly important. I would encourage the authors to at least consider talking about this in the Future Directions part of the manuscript. When parts of ENSO are in a particular phase, what does it tell us about the terrestrial carbon store response, should general climate warming be in that state in a persistent way?

In the "Future Directions", the authors note that a more formal use of multiple DGVMs will help. The paper does not consider future projections, and it would certainly be interesting to see Figure 2d,e,f extended under the CMIP5/6 ensemble, maybe in a follow-on paper. Assessment of future findings will also have to be related to how well individual ESMs performing in projecting ENSO characteristics. The authors could also provide a couple of sentences on how others might be encouraged by this analysis to use data to assess the carbon cycle components of their analyses. Datasets do exist of the carbon cycle components, and for instance of NPP ("MODIS NPP"?). While some gridded datasets of terrestrial carbon do contain aspects of models in them e.g. to disaggregate from point to all locations, they still remain highly useful guides and are still "measurements" as such. What would comparisons show between the model-based findings of this paper and terrestrial carbon cycling measurements? The authors could then discuss in a short paragraph how data can constrain which aspects of land surface responses are performing well, and where there are deficiencies. Once constrained, the implications under future climates can be characterised. Although ecosystem acclimation effects might have to be accounted for, this would still offer an extra way to use current interannual variability to tell us about climate impacts. That

is the variations might tell us terrestrial carbon cycle response under a permanently adjusted near-surface climatic state. This paper provides a framework of which ENSO "expressions" to focus on, on the path to constraining future projections of land carbon cycle change.

The paper includes a particularly good introduction, and the broad literature search is undertaken well, capturing all the main recent papers on ENSO-Carbon cycle teleconnections.

I am happy to see any new paper version, and I will try and return any further comments much more promptly.

Small things

The word "expression" is used quite a bit e.g. in the discussion of the Central-Pacific and Eastern-Pacific features of ElNino. "Attributes" or "features" may be better words?

Can the diagrams could be tidied up a little more? To my eyes at least, some of the features of – for instance – Figure 2 are difficult to see. Slightly thicker curve linewidths might help, and without obscuring each other.

A better use of the colourbars would help in Figure B1 for instance, to understand better the geographical spread. To achieve this could be by including colour steps that are not all of identical amounts. Clustering of some colour bounds more around the zero value will reveal more information in the maps?

---

## Author Comment (AC2) · 30 Nov 2020

We thank the reviewer for taking the time to review our manuscript and provide constructive comments.

In revising our manuscript, we noted that our model simulations had used a fixed pre-industrial nitrogen deposition rate. In our resubmission, we reflected that it would make more sense to show results from LPJ-GUESS with the nitrogen cycle switched off. This was because the principal aim of our paper was to explore the sensitivity of the carbon cycle to 'expressions' of El Nino and we might expect that this sensitivity would be greatest using the C-only version of LPJ-GUESS as carbon uptake is not limited by nutrient availability (which may decline with water availability in dry years, when nitrogen immobilisation rates increase). Nevertheless, as one of our main regions of interest was the tropics, we would not expect a strong limitation by nitrogen (Vitousek et al. 1984) and as a result, we do not anticipate a strong sensitivity in our results to our choice of biogeochemical cycle. To assure the Editor/Reviewer of this insensitivity we have shown the results of both cycles (N-cycle on/off) below (Fig 1). We also used this opportunity to update the model comparison against the more recent TRENDY v7 runs.

Overall, we found that LPJ-GUESS is close to the TRENDY v7 ensemble mean and simulations are mostly within the model range (i.e. across TRENDY models) when we switch the nitrogen cycle off. The spatial distribution of the summed composite GPP anomalies (see fig. 2) shows that LPJ-GUESS picks up the main anomalies associated with EP El Nino events and remains within the TRENDY models' range. Finally, LPJ-GUESS has a strong negative bias in Australia. As our results show, Australia does not make a large contribution to long-term changes in any of the carbon fluxes and pools.

[Figure]

Fig. 1. Monthly composite anomalies during the El Nino developing (y0) and decaying (y1) year in gross primary production (GPP; green lines) and terrestrial ecosystem

respiration (TER; sum of autotrophic and heterotrophic respiration; red lines) for all CP and EP El Nino events listed in appendix table A1 averaged over the globe, the tropics (23°S–23°N) and Australia. The dotted lines show the TRENDY v7 composite, the solid lines are the individual LPJ-GUESS run where we switch of the nitrogen cycle, the dashed lines show the model runs with dynamic nitrogen cycling (compare Wang et al., 2018). ρ and R² show the correlation coefficients and R² values between the LPJ-GUESS and the TRENDY ensemble mean. The shaded area shows the model spread of the individual TRENDY models.

[Figure]

Fig. 2: Composite anomalies in gross primary production (GPP) summed over the the El Nino developing and decaying year for all CP and EP El Niño events listed in tab. B1 for the individual TRENDY models, the TRENDY composite and the individual LPJ–GUESS run (compare Wang et al., 2018).

Below we address the reviewer's comments point by point. We add our replies in italics and highlight suggested modifications in the manuscript in red.

Referee #2

In this manuscript, Teckentrup et al. used LPJ-GUESS forced by manipulated climate datasets to study the influences of two expressions of El Nino (CP and EP) on the terrestrial carbon cycle. The authors suggested that the expressions of El Nino only influence interannual variability of NBP (e.g. CP caused larger IAV in NBP than EP at the global scale) but not the long-term change in NBP. They concluded that the relative frequency of CP and EP is not critical in models as CP/EP did not yield detectable changes in long-term NBP. The science question is interesting, the story is well told and there is no major flaw in the method. That being said, there are a few questions that puzzled me after reading the manuscript, which I hope the authors could clarify a bit before I could support it.

*We thank the reviewer for their assessment and the acknowledgement of our contributions.*

1. One of the novel points presented is that "impact (of CP and EP) on longer timescales is not well understood". El Nino, either CP or EP, is known to dominate the interannual variability of terrestrial carbon cycling. It is not clearly stated in the Introduction why we would expect an influence of CP/EP El Nino at longer timescales in the first place. In another word, would it be a surprise that CP/EP El Nino exert no change on long-term NBP, as we already known that El Nino influences IAV rather than long-term variability of the carbon cycle. Perhaps the relative more frequent CP occurrences in the future could be an issue long term but the current models may not include proper mechanisms (i.e. shift in species composition, acclimations) to answer the question.

   *We agree that El Nino studies have mostly focused on interannual timescales. However, in a recent study, Park et al. 2020 found that decadal variability in ENSO influences the long term terrestrial global carbon cycle. Further, as noted by the reviewer, a shift in El Nino patterns could alter cumulative net biome production, which may alter competitive patterns of plant species, both of which may influence the carbon stored in vegetation and soil. Similarly, interannual variability in precipitation patterns induced by different types of El Nino might change vegetation dynamics in semiarid areas/savanna ecosystems. As a result, we do think the focus of our study is warranted.*

   *Our results imply that the expression of El Nino did not lead to any of the changes described in earlier studies. We agree with the reviewer that it is possible that this may in part relate to missing mechanisms that would capture species composition changes. Yet we think this is unlikely to be the explanation, given that El Nino events are very short-lived and spatially variable which likely prevents a direct shift in vegetation in most biomes due to changes in meteorology. Whilst this summary of our findings agrees with the reviewer's point above, we still had to do the exploratory work to determine whether this was in fact true.*

   *We have amended the motivation text in the introduction to more clearly capture these issues:*

*'A shift in El Nino patterns could change cumulative net biome production, which may alter competitive patterns of plant species, both of which may influence the carbon stored in vegetation and soil. Similarly, interannual variability in precipitation patterns induced by different types of El Nino might change vegetation dynamics in semiarid areas/savanna ecosystems (cf. Scheiter and Higgins, 2009; Whitley et al., 2017).'*

2. The study is aimed at studying the sensitivity of the terrestrial carbon cycle to CP/EP El Nino. And the author did so by replacing the climate anomalies during CP to EP and vice versa. CP is reported to cause larger global IAV than EP.

   *Yes – that is correct.*

   My concerns is: (using global simulation as an example) is this larger sensitivity of the terrestrial carbon cycle to CP is due to the changes in the inherent climate sensitivity of carbon during CP/EP, or is this simply caused by the generally larger climate anomalies during CP (Fig. B5). I would assume the reason is the latter, as the inherent climate sensitivity of carbon cycle is essentially predefined by the model (in this case LPJ-GUESS) structure, so what we see here (IAV of NBP in CP > EP) is perhaps just because the IAV of climate in CP > EP.

   *We agree with the reviewer that the model simulations suggest that the results largely follow the climate anomalies. Employing a DGVM provides the opportunity to explore possible lag effects, such as changes in fire dynamics, and changes vegetation composition. These lag processes that might be captured by a DGVM allowed us to explore the reviewer's question about the 'inherent climate sensitivity of carbon during CP/EP'. In fact, we found that the perturbations forced on the vegetation were too small to cause significant carryover effects, and conclude therefore that climate anomalies were the key control for the observed changes. As with the reviewer's previous point about 'long-term responses', we do not think the answer is self-evident and felt it was important to explore the model sensitivities.*

3. missed chance on the spatial and phenology of carbon fluxes. While I have doubts about the reported difference between CP and EP at interannual or longer time scales, I feel their difference is perhaps more pronounced at seasonal scales and spatial, when CP and EP show apparent contrasting temporal patterns (e.g. Fig 1). As was also noted by Chylek et al. 2018, the time delay of CO2 rise after SST increase is one of the pronounced differences, and the difference is only around 3 months. Focusing on longer time scales might easily just averaged out these important characteristics. I think the authors have done a nice job in demonstrating the spatial difference of carbon sinks under CP/EP, and these results perhaps worth more highlights.

   *We agree with the reviewer that changes are likely more significant at seasonal timescales. As the reviewer notes, our results show that even if there are strong impacts on shorter timescales, these effects disappear on decadal timescales which*

*is a key conclusion in our paper. We therefore choose not to further explore shorter timescales given this is already something that has been reported in the literature.*

4. With that, I would also say it maybe a stretch to say CP/EP is not critical in future models, as their major difference is likely to be clearer seasonally and spatially (e.g. different carbon sink distribution, phenology of carbon uptake).

   *We agree with the reviewer that the different expressions of El Nino affect the terrestrial carbon cycle on interannual timescales and that individual events potentially have large impact on specific regions. However, our results suggest that the perturbations linked to the expression of El Nino might be too small to trigger changes in vegetation dynamics that last longer than a season or a year. Nevertheless, we have adjusted the future directions:*

   *'Based on this analysis we suggest that our model sensitivity would likely be similar to that displayed by the other TRENDY models, although we would anticipate subtle regional differences, particular in the tropics if an alternative DGVM had been used'*

   *and the conclusions:*

   *'Our results therefore suggest that the impact of different expressions of El Nino on the carbon cycle on long time scales is likely to be small.'*

5. L11. Please specify what kind of longer time scale effect (i.e. decadal mean, decadal variation or trend?)

   *We used 'longer time scale effect' to describe the effect a climate with only CP El Nino or only EP El Nino events might have on terrestrial vegetation after 45 years. We do not analyse decadal mean, variation or trend but rather assess the effect by comparing the final year of the two different scenarios to that of the control run (where both expressions of El Nino occur).*

6. L84 and L104. If CRU-NCEP v7 covers 1901-2016, why not consider the 2015/2016 El Nino in the analysis.

   *We chose the year 2013 as the last year of our experiment run because it is ENSO-neutral.*

7. L84. By saying CRU, did you mean CRUNCEP.

   *Thank you, yes and we changed the text accordingly.*

8. L119-120. I am not sure I understand how to choose the replacements for CP and EP correctly. Why there is a need to resample climate anomalies using ONI and how do we locate the CP that is used to replace a EP (in the same 10-year window shown in Fig 1?).

*We use the approach according to Yu and Kim, 2009. They use the ONI index to identify El Nino events which comprise both CP and EP El Nino events. Based on four different indices, they then further differentiate between CP and EP El Nino events. We changed the description in the 'Identification of El Nino events' section to:*

*'They first classified El Nino events based on the Oceanic Nino Index (ONI) which comprise both CP and EP El Nino events. Based on four indices, they then further differentiate between CP and EP El Nino events.'*

*We also use the ONI index as a guidance for the replacement of the individual El Nino events. We replaced an EP El Nino event with a CP type (and vice versa), when both events start, end and peak at the similar times in the year according to the ONI index and have similar magnitudes in the ONI index (see methods).*
*We updated the methods section:*

*'We used the ONI index to define the start, end and strength of the individual El Nino events and resampled the climate anomalies based on the ONI. We replaced anomalies in the climate forcing associated with El Nino events according to the best fit in duration and amplitude in ONI, i.e. events that start and end at a similar time in the year and have a similar timing and magnitude of the peak in ONI.'*

9. L210. Does LPJ-GUESS have a component to simulate species composition?

   *Thanks for pointing out this potential confusion. LPJ-GUESS represents vegetation in form of plant functional types, not individual species. We replaced 'species composition' with 'vegetation composition'.*

10. B1-B4: Unit of carbon fluxes in supplementary figures. Per m2?

    *Thank you for pointing this out, we changed it accordingly and updated the figures.*

Scheiter, S. and Higgins, S.I. (2009), Impacts of climate change on the vegetation of Africa: an adaptive dynamic vegetation modelling approach. Global Change Biology, 15: 2224-2246. doi:10.1111/j.1365-2486.2008.01838.x

Yu, J.-Y. and Kim, S. T.: Identifying the types of major El Niño events since 1870, International Journal of Climatology, 33, 2105 2112,465https://doi.org/10.1002/joc.3575, https://rmets.onlinelibrary.wiley.com/doi/abs/10.1002/joc.3575, 2013.

Vitousek, P. M. Litterfall, nutrient cycling, and nutrient limitation in tropical forests. Ecology 65, 285–298 (1984).

Whitley, R., Beringer, J., Hutley, L. B., Abramowitz, G., De Kauwe, M. G., Evans, B., Haverd, V., Li, L., Moore, C., Ryu, Y., Scheiter, S., Schymanski, S. J., Smith, B., Wang, Y.-P., Williams, M., and Yu, Q.: Challenges and opportunities in land surface modelling of savanna ecosystems, Biogeosciences, 14, 4711–4732, https://doi.org/10.5194/bg-14-4711-2017, 2017.

---

## Author Response (AR2)

Dear editor and referees,

In revising our manuscript, we noted that our model simulations had used a fixed pre-industrial nitrogen deposition rate. In our resubmission, we reflected that it would make more sense to show results from LPJ-GUESS with the nitrogen cycle switched off. This was because the principal aim of our paper was to explore the sensitivity of the carbon cycle to 'expressions' of El Nino and we might expect that this sensitivity would be greatest using the C-only version of LPJ-GUESS as carbon uptake is not limited by nutrient availability (which may decline with water availability in dry years, when nitrogen immobilisation rates increase). Nevertheless, as one of our main regions of interest was the tropics, we would not expect a strong limitation by nitrogen (Vitousek et al. 1984) and as a result, we do not anticipate a strong sensitivity in our results to our choice of biogeochemical cycle. To assure the Editor/Reviewer of this insensitivity we have shown the results of both cycles (N-cycle on/off) below (Fig 1). We also used this opportunity to update the model comparison against the more recent TRENDY v 7 runs.

Overall, we found that LPJ-GUESS is close to the TRENDY v7 ensemble mean and simulations are within the model range (i.e. across TRENDY models) when we switch the nitrogen cycle off. The spatial distribution of the summed composite GPP anomalies (see fig. 2) shows that LPJ-GUESS picks up the main anomalies associated with EP El Nino events and remains within the TRENDY models' range. Finally, LPJ-GUESS has a strong negative bias in Australia. As our results show, Australia does not make a large contribution to long-term changes in any of the carbon fluxes and pools.

[Figure]

Fig. 1. Monthly composite anomalies during the El Nino developing (y0) and decaying (y1) year in gross primary production (GPP; green lines) and terrestrial ecosystem respiration (TER; sum of autotrophic and heterotrophic respiration; red lines) for all CP

and EP El Nino events listed in appendix table A1 averaged over the globe, the tropics (23°S–23°N) and Australia. The dotted lines show the TRENDY v7 composite, the solid lines are the individual LPJ-GUESS run where we switch of the nitrogen cycle, the dashed lines show the model runs with dynamic nitrogen cycling (compare Wang et al., 2018). ρ and $R^2$ show the correlation coefficients and $R^2$ values between the LPJ-GUESS and the TRENDY ensemble mean. The shaded area shows the model spread of the individual TRENDY models.

[Figure]

Fig. 2: Composite anomalies in gross primary production (GPP) summed over the the El Nino developing and decaying year for all CP and EP El Niño events listed in appendix table B1 for the individual TRENDY models, the TRENDY composite and the individual LPJ–GUESS run (compare Wang et al., 2018).

Below we address the editor's and reviewers' comments point by point. We add our replies in italics and highlight suggested modifications in the manuscript in red.

*Response to the editor*

*We thank the editor for taking the time to review our manuscript and provide constructive comments.*

1. Dear authors,

   Thank you for uploading responses to the referees. As you have seen several of the reviewers were enthusiastic about the value of your analysis and the reporting of this negative result. I also appreciate the completeness in re-running your analyses with a different nitrogen cycle parametrization. At the same time, Reviewer 2 had a number of major concerns about the way the motivation for the work is presented,

   *Our general motivation was to examine the sensitivity of the longterm dynamics of the terrestrial carbon cycle to a change in the frequency of CP- and EP-El-Nino-events. In our revised paper, we now add further details on the general motivation in the introduction:*

   *'A shift in El Nino patterns could change cumulative net biome production, which may alter competitive patterns of plant functional types, both of which may influence the carbon stored in vegetation and soil (e.g. Park et al., 2020). Similarly, interannual variability in precipitation patterns induced by different types of El Nino might result in a shift in vegetation distributions, particular at climatic transition zones, or in water-limited environments, e.g. semi-arid areas/savanna ecosystems (cf. Scheiter and Higgins, 2009; Whitley et al., 2017).'*

   *We also included more detail on the reasons for exploring our question with a DGVM:*

   *'In our study we used a dynamic global vegetation model (DGVM) to examine the sensitivity of the terrestrial carbon cycle to changes in El Nino patterns. In response to climate, DGVMs predict global vegetation distributions based on plant physiology, competition, demography and vegetation structure (Sitch et al. 2003; Woodward and Lomas 2004). In particular, these models also consider how fire dynamics and vegetation composition may respond to a shift in climate. In the past DGVMs have been widely used to examine how vegetation distributions may change in response to climate (Hickler et al., 2012; Martens et al., 2020) and fire (Kelley et al., 2014).'*

   *We also further discussed the shortcomings of DGVMs*

   *'Similarly, models differ in their sensitivity of the carbon cycle as water becomes limiting (Powell et al., 2013), which may affect the magnitude of carbon uptake in extreme El Nino years. Fisher et al. (2018) also highlighted hydrodynamics as well as the representation of demographic processes (e.g. recruitment and mortality) and fire disturbance as areas of uncertainty and promising for model development.'*

2. and Reviewer 3 also had significant suggestions for framing your work's implications in terms of differences between your ENSO-expression findings and other work suggesting that interannual variability can affect long-term outcomes.

*We carefully considered R3 suggestions and have revised the text accordingly. First, we add in the discussion:*

*'Interconnections between the terrestrial carbon cycle and ENSO have been widely explored (e.g. Zhang et al., 2018; Rödenbeck et al., 2018; Chylek et al., 2018). Whilst we also find key NBP variability on annual to decadal timescales (see fig. 2), particularly in CP years (accumulated NBP = 7.7 PgC), we did not find that this shorter timescale variability translated into sustained trends (1968-2013) in ecosystem fluxes, or shifts in vegetation distributions (see fig. 3).'*

*We now further critique how robust our results are and add in the future directions:*

*'The spatial distribution of the composite anomalies shows that LPJ-GUESS captures the features of anomalies in GPP associated with EP El Nino events compared to the individual models and the TRENDY model ensemble (see appendix figure B11). In contrast, LPJ-GUESS generally simulates weaker anomalies in GPP associated with CP El Nino events in Brazil and Western Africa compared to the ensemble mean and most individual models. This low sensitivity might also explain the relatively low correlation and R2 values in appendix figure B10 for tropical regions and may dampen the overall response to the CP-only-scenario. We note however that LPJ-GUESS still is within the model range and can therefore be viewed as representative. In addition, LPJ-GUESS has a strong negative bias in Australia. As our results show, Australia does not make a large contribution to long-term changes in any of the carbon fluxes and pools. We also examined the sensitivity of our results to the use of a nitrogen cycle with LPJ-GUESS (see appendix figure B10) but did not find a strong sensitivity, most likely because nitrogen is not thought to be strongly limiting in the tropics (Vitousek et al., 1984).'*

*We have added text to discuss approaches to extend our research question into the future:*

*'Moreover, exploring the impact of different expressions of El Nino in a future climate would be worthwhile. However, we note that this would probably require multiple DGVMs to account for the uncertainty associated with the vegetation responses to [CO$_2$] and interactions with nutrients (Zaehle et al., 2014). In addition, the representation of ENSO diversity in CMIP5 and CMIP6 models is highly uncertain due to model biases, especially in the equatorial Pacific, resulting in low model agreement (e.g. Freund et al., 2020). Therefore, to obtain robust results, a future experimental design would also require an ensemble of climate forcing input datasets.'*

*We also added text about extending our work with validation against satellite data:*

*'Lastly, a comparison with satellite-derived observations might help to estimate whether LPJ-GUESS or indeed an alternative DGVM, captures the correct sensitivity in the response of vegetation dynamics to ENSO events. Nevertheless, as direct global measurements of carbon fluxes do not exist, and those that do are often based on models themselves, future work might restrict comparison to less direct proxies of variability e.g. leaf area index (Zhu et al., 2013) and/or GRACE terrestrial water storage (Rodell et al., 2004).'*

*Finally, we revisited and softened our statement about the implication of our work acknowledging the problem of using only one DGVM*

*'Based on this analysis, we suggest that our model sensitivity would likely be similar to that displayed by the other TRENDY models, although we would anticipate subtle regional differences, particular in the tropics if an alternative DGVM had been used. Especially for EP El Nino events, LPJ-GUESS diverges from the TRENDY ensemble mean that cannot be explained by nutrient limitation and suggests a different sensitivity to the meteorological drivers (see appendix figure B10).'*

*As well as changing our overall conclusion:*

*'Our results therefore suggest that the impact of different expressions of El Nino on the carbon cycle on long timescales is likely to be small.'*

In light of these comments, major revisions to the writing are needed. While I appreciate many of your posted responses to the referees that the concerns raised are not likely to qualitatively change the take-away results of your manuscript, it would still be useful to amend the manuscript writing/discussion to make sure that these issues are as clear to the reader as you make them for the reviewers. For example, the Park et al 2020 motivating reference is not proposed to be cited, the lagged effects mentioned in response to reviewer 2 will not be mentioned, etc. Indeed comments 2 and 3 from Reviewer 3 are not proposed to be addressed in the manuscript text at all. Please reconsider some of these before submitting a revised manuscript.

We further included the corrections suggested by Referee 1.

(1) In abstract, "as well as different lags in terrestrial CO2 release to the atmosphere following increased tropical near surface air temperature". Till now, the dominant driver (temperature vs precipitation) remains debatable. So it is more reasonable to modify it as "... following increased tropical near surface air temperature and decreased precipitation".

*Thank you for the suggestion, we changed the abstract accordingly.*

(2) In Results P6L154: "Overall, global changes in NBP accumulate to 9.6 PgC and 4.5 PgC for the CP– and the EP–only–scenario, respectively". I am a little confused that both of the CP and EP - only-scenario contribute to increase the NBP accumulation relative to the control run (mixed events). Do you have any

explain why the control (mixed CP and EP) have the lowest cumulative NBP (Fig. 2d)? Is it related to the method of the manipulation of climate data?

*The lower cumulative NBP in the control relative to CP/EP is as the reviewer notes an emergent feature of the climate manipulation combined with the inherent stochasticity (e.g. growth, fire dynamics) in LPJ-GUESS, i.e. the result is by chance.*

(3) P10L223-224: the study of Park et al (2020) tells a different story that they focus on the decadal variability. It seems you can delete it here because your study is totally different.

*Thank you for pointing this out, we deleted the reference in the discussion.*

(4) This study is based on an assumption that change of the expression of El Nino may not influence the El Nino-La Nina cycle. But actually, if EP is replaced by CP or vice versa, it may influence the following La Nina event. So add sentences in the discussion to illustrate this potential limitation.

*We now include in the discussion:*

*We further neglect possible interactions between consecutive ENSO events. For example, strong El Nino events tend to peak in the eastern Pacific, and these tend to be followed by a La Nina event. However, the influence of a preceding El Nino on the characteristics of the La Nina event is not clear (Santoso et al., 2017).*

*Response to Referee #1*

*We thank the reviewer for taking the time to review our manuscript and provide constructive comments.*

This manuscript investigates the impacts of different expressions of El Nino on the long-term terrestrial carbon storages, using a DGVM LPJ-GUESS with the manipulated climate forcing. They pointed out that CP and EP events can significantly influence the interannual variability of terrestrial carbon cycle, but cannot lead to NBP trend. Therefore, they suggest that future simulations of carbon cycle may not need to well simulate the expressions of El Ninos in Earth System model. The method is well described and writing is clear with concise and clear conclusions.

*We thank the reviewer for their assessment and the acknowledgement of our contributions.*

3. L120: "associated with El Nino events according to the best fit in duration and amplitude in ONI...". Because there are actually 6 CP, 7 EP, and 2 Mix, you can clearly show the replacement relationships in the table for the manipulations (like in Table A1). It can be more straightforward for us to understand it.

   *We added two more columns in the appendix table A1 where we specify the replacement relationships.*

4. The units in spatial patterns in Figure B1–B4 are not correct. For example, flux is gC=>gC/m2/yr?, carbon pool is gC yr-1?=>gC?

   *Thank you for pointing this out, we changed it accordingly and updated the figures.*

5. In Discussion: Some aspects can be mentioned further.
   a) ENSO diversity (Capotondi et al., 2015): Although replace the CP and EP events based on their durations and amplitudes, every ENSO event is unique with different spatial impacts.

   *We agree that every El Nino or La Nina event is unique. We did mention this as a limitation in the discussion, but have now included the Capotondi et al., 2015 citation too:*

   *'Individual El Nino events vary in location, timing and magnitude (e.g. Capotondi et al., 2015) and teleconnections are influenced by the background climate and climate variability (e.g. the Indian Ocean Dipole).'*

   b) Changes in frequency of ENSO occurrence in future: Though it maybe doesn't influence your conclusions, you can discuss that frequency change may have some influences.

   *We agree that work that revisits this question for the perspective of a future climate may well be warranted. However, there is little evidence suggesting that the frequency of El Nino – La Nina cycles might change in the future. Some*

*studies indicate changes in the properties of El Nino events, i.e. magnitude (e.g. Wang et al., 2019) as well as spatial features (e.g. Yeh et al., 2009). However, the representation of ENSO diversity in CMIP5 and CMIP6 models is associated with high uncertainty due to model biases especially in the equatorial Pacific, resulting in low model agreement (e.g. Freund et al., 2020). A future experiment set-up would need an ensemble of climate forcing datasets and probably multiple DGVMs since the results may be very sensitive to assumptions related to vegetation responses to $[CO_2]$ and interactions with nutrients (Zaehle et al., 2014).*

*These are important issues and therefore we added the following into the future directions section:*

*'Moreover, exploring the impact of different expressions of El Nino in a future climate would be worthwhile. However, we note that this would probably require multiple DGVMs to account for the uncertainty associated with the vegetation responses to $[CO_2]$ and interactions with nutrients (Zaehle et al., 2014). In addition, the representation of ENSO diversity in CMIP5 and CMIP6 models is highly uncertain due to model biases, especially in the equatorial Pacific, resulting in low model agreement (e.g. Freund et al., 2020). Therefore, to obtain robust results, a future experimental design would also require an ensemble of climate forcing input datasets.'*

*Response to Referee #2*

In this manuscript, Teckentrup et al. used LPJ-GUESS forced by manipulated climate datasets to study the influences of two expressions of El Nino (CP and EP) on the terrestrial carbon cycle. The authors suggested that the expressions of El Nino only influence interannual variability of NBP (e.g. CP caused larger IAV in NBP than EP at the global scale) but not the long-term change in NBP. They concluded that the relative frequency of CP and EP is not critical in models as CP/EP did not yield detectable changes in long-term NBP. The science question is interesting, the story is well told and there is no major flaw in the method. That being said, there are a few questions that puzzled me after reading the manuscript, which I hope the authors could clarify a bit before I could support it.

*We thank the reviewer for their assessment and the acknowledgement of our contributions.*

6. One of the novel points presented is that "impact (of CP and EP) on longer timescales is not well understood". El Nino, either CP or EP, is known to dominate the interannual variability of terrestrial carbon cycling. It is not clearly stated in the Introduction why we would expect an influence of CP/EP El Nino at longer timescales in the first place. In another word, would it be a surprise that CP/EP El Nino exert no change on long-term NBP, as we already known that El Nino influences IAV rather than long-term variability of the carbon cycle.

   *We agree that El Nino studies have mostly focused on interannual timescales. However, in a recent study, Park et al. 2020 found that decadal variability in ENSO influences the long term terrestrial global carbon cycle through changes in climate which in turn affect the vegetation's carbon uptake and growth patterns. Further, as noted by the reviewer, a shift in El Nino patterns could alter cumulative net biome production, which may alter competitive patterns of plant species, both of which may influence the longer term carbon stored in vegetation and soil. Similarly, interannual variability in precipitation patterns induced by different types of El Nino might change vegetation dynamics in semiarid areas/savanna ecosystems. As a result, we do think the focus of our study was warranted.*

   *We have amended the motivation text in the introduction to more clearly capture these issues:*

   *'A shift in El Nino patterns could change cumulative net biome production, which may alter competitive patterns of plant functional types, both of which may influence the carbon stored in vegetation and soil (e.g. Park et al., 2020). Similarly, interannual variability in precipitation patterns induced by different types of El Nino might result in a shift in vegetation distributions, particular at climatic transition zones, or in water-limited environments, e.g. semi-arid areas/savanna ecosystems (cf. Scheiter and Higgins, 2009; Whitley et al., 2017).'*

7. Perhaps the relative more frequent CP occurrences in the future could be an issue long term but the current models may not include proper mechanisms (i.e. shift in species composition, acclimations) to answer the question.

*We agree with the reviewer that it is possible that this may in part relate to missing mechanisms that would capture vegetation composition changes. We discuss in the future directions that the use of a single model does not allow to quantify uncertainties associated with alternative models and/or missing processes:*

*'Since we only use a single model we cannot quantify uncertainties associated with alternative models and/or missing processes.'*

*We further explicitly mention that not including acclimation might lead to an overestimation of carbon sensitivity to temperature changes on short time scales:*

*'For example, LPJ-GUESS, similar to many land surface and dynamic global vegetation models, does not account for acclimation of plant respiration to increased temperature, and may consequently overestimate the carbon sensitivity to temperature changes on short timescales (e.g. Wang et al., 2020; Huntingford et al., 2017; Smith et al., 2015).'*

*In addition, we now include in the future directions*

*'Similarly, models differ in their sensitivity of the carbon cycle as water becomes limiting (Powell et al., 2013), which may affect the magnitude of carbon uptake in extreme El Nino years. Fisher et al. (2018) also highlighted hydrodynamics as well as the representation of demographic processes (e.g. recruitment and mortality) and fire disturbance as areas of uncertainty and promising for model development.'*

*However, we think the lack of process presentation is unlikely to be the explanation, given that El Nino events are very short-lived and spatially variable which likely prevents a direct shift in vegetation in most biomes due to changes in meteorology. Whilst this summary of our findings agrees with the reviewer's point above, by undertaking this study we were able to demonstrate it to be true.*

8.  The study is aimed at studying the sensitivity of the terrestrial carbon cycle to CP/EP El Nino. And the author did so by replacing the climate anomalies during CP to EP and vice versa. CP is reported to cause larger global IAV than EP.

    *Yes – that is correct.*

9.  My concerns is: (using global simulation as an example) is this larger sensitivity of the terrestrial carbon cycle to CP is due to the changes in the inherent climate sensitivity of carbon during CP/EP, or is this simply caused by the generally larger climate anomalies during CP (Fig. B5). I would assume the reason is the latter, as the inherent climate sensitivity of carbon cycle is essentially predefined by the model (in this case LPJ-GUESS) structure, so what we see here (IAV of NBP in CP > EP) is perhaps just because the IAV of climate in CP > EP.

    *We agree with the reviewer that the model simulations suggest that the results largely follow the climate anomalies. However, employing a DGVM provides the opportunity to also explore possible lag effects in response to CP/EP events, such as*

*changes in fire dynamics, and changes vegetation structure/composition. We adjusted the future directions to read:*

*'In our study we used a dynamic global vegetation model (DGVM) to examine the sensitivity of the terrestrial carbon cycle to changes in El Nino patterns. In response to climate, DGVMs predict global vegetation distributions based on plant physiology, competition, demography and vegetation structure (Sitch et al. 2003; Woodward and Lomas 2004). In particular, these models also consider how fire dynamics and vegetation composition may respond to a shift in climate. In the past DGVMs have been widely used to examine how vegetation distributions may change in response to climate (Hickler et al., 2012; Martens et al., 2020) and fire (Kelley et al., 2014).'*

*These lag processes that might be captured by a DGVM allowed us to explore the reviewer's question about the 'inherent climate sensitivity of carbon during CP/EP'. In fact, we found that the perturbations forced on the vegetation were too small to cause significant carryover effects, and conclude therefore that climate anomalies were the key control for the observed changes. We address this in the discussion:*

*'Overall, the high spatial and temporal variability in the changes suggest that the effect of different expressions of El Nino on the terrestrial carbon cycle are important for predicting responses on interannual timescales (e.g. the atmospheric CO2 growth rate) but are unlikely to affect the terrestrial carbon balance on longer timescales. Our model results imply that the anomaly patterns in the El Nino expression on climate forcing were too variable (and short-lived) to result in systematic shifts in vegetation composition.'*

*As with the reviewer's previous point about 'long-term responses', we do not think the answer was self-evident and felt it was important to explore the model sensitivities.*

10. missed chance on the spatial and phenology of carbon fluxes. While I have doubts about the reported difference between CP and EP at interannual or longer time scales, I feel their difference is perhaps more pronounced at seasonal scales and spatial, when CP and EP show apparent contrasting temporal patterns (e.g. Fig 1). As was also noted by Chylek et al. 2018, the time delay of CO2 rise after SST increase is one of the pronounced differences, and the difference is only around 3 months. Focusing on longer time scales might easily just averaged out these important characteristics. I think the authors have done a nice job in demonstrating the spatial difference of carbon sinks under CP/EP, and these results perhaps worth more highlights.

*We agree with the reviewer that changes are likely to be more significant at seasonal timescales as was examined previously (e.g. Chylek et al. 2018) or on decadal timescales (Park et al. 2020). Here, we instead focussed on the accumulation of these sub-annual responses to determine their impact (or lack thereof) on decade timescales. As the reviewer notes, our results show that even if there are strong impacts on shorter timescales, these effects disappear on decadal timescales which is a key conclusion in our paper. This is important in estimating*

*longer timescale carbon sinks because had these shorter timescale carbon variability accumulated, this would have meant the CP/EP variability was important, with consequences for modelling El Nino.*

*We have also extended the discussion:*

*'Interconnections between the terrestrial carbon cycle and ENSO have been widely explored (e.g. Zhang et al., 2018; Rödenbeck et al., 2018; Chylek et al., 2018). Whilst we also find key NBP variability on annual to decadal timescales (see fig. 2), particularly in CP years (accumulated NBP = 7.7 PgC), we did not find that this shorter timescale variability translated into sustained trends (1968-2013) in ecosystem fluxes, or shifts in vegetation distributions (see fig. 3).'*

11. With that, I would also say it maybe a stretch to say CP/EP is not critical in future models, as their major difference is likely to be clearer seasonally and spatially (e.g. different carbon sink distribution, phenology of carbon uptake).

    *We agree with the reviewer that the different expressions of El Nino affect the terrestrial carbon cycle on interannual timescales and that individual events potentially have large impact on specific regions. However, our results suggest that the perturbations linked to the expression of El Nino might be too small to trigger changes in vegetation dynamics that last longer than a season or a year. Nevertheless, we have adjusted the future directions:*

    *'Based on this analysis, we suggest that our model sensitivity would likely be similar to that displayed by the other TRENDY models, although we would anticipate subtle regional differences, particular in the tropics if an alternative DGVM had been used. Especially for EP El Nino events, LPJ-GUESS diverges from the TRENDY ensemble mean that cannot be explained by nutrient limitation and suggests a different sensitivity to the meteorological drivers (see appendix figure B10).'*

    *and the conclusions:*

    *'Our results therefore suggest that the impact of different expressions of El Nino on the carbon cycle on long timescales is likely to be small.'*

12. L11. Please specify what kind of longer time scale effect (i.e. decadal mean, decadal variation or trend?)

    *We used 'longer time scale effect' to describe the effect a climate with only CP El Nino or only EP El Nino events might have on terrestrial vegetation after 45 years. We do not analyse decadal mean, variation or trend but rather assess the effect by comparing the final year of the two different scenarios to that of the control run (where both expressions of El Nino occur).*

    *We have now added to the discussion:*

*'We analyse the effect that a climate with only CP El Nino or only EP El Nino events might have on terrestrial vegetation after 45 years by comparing the final year of the two different scenarios to that of the control run (where both expressions of El Nino occur).'*

13. L84 and L104. If CRU-NCEP v7 covers 1901-2016, why not consider the 2015/2016 El Nino in the analysis.

    *The version of CRU-NCEP we had on our system when we carried out the analysis and wrote the manuscript only extended to 2015. To allow us to compare our across experiments we need our experiments to end with an ENSO-neutral year, i.e. 2013, when we ended our analysis. So, even if we had access to the data from 2015/16, we would not have used it within our experimental framework because we need to end with an ENSO-neutral year.*

14. L84. By saying CRU, did you mean CRUNCEP.

    *Thank you, yes and we changed the text accordingly.*

15. L119-120. I am not sure I understand how to choose the replacements for CP and EP correctly. Why there is a need to resample climate anomalies using ONI and how do we locate the CP that is used to replace a EP (in the same 10-year window shown in Fig 1?).

    *We use the approach according to Yu and Kim, 2009. They use the ONI index to identify El Nino events which comprise both CP and EP El Nino events. Based on four different indices they then further differentiate between CP and EP El Nino events. We changed the description in the 'Identification of El Nino events' section to:*

    *'They first classified El Nino events based on the Oceanic Nino Index (ONI) which comprise both CP and EP El Nino events. Based on four indices, they then further differentiate between CP and EP El Nino events.'*

    *We also use the ONI index as a guidance for the replacement of the individual El Nino events. We replaced an EP El Nino event with a CP type (and vice versa), when both events start, end and peak at the similar times in the year according to the ONI index and have similar magnitudes in the ONI index (see methods).*
    *We updated the methods section:*

    *'We used the ONI index to define the start, end and strength of the individual El Nino events and resampled the climate anomalies based on the ONI. We replaced anomalies in the climate forcing associated with El Nino events according to the best fit in duration and amplitude in ONI, i.e. events that start and end at a similar time in the year and have a similar timing and magnitude of the peak in ONI.'*

16. L210. Does LPJ-GUESS have a component to simulate species composition?

*Thanks for pointing out this potential confusion. LPJ-GUESS represents vegetation in form of plant functional types, not individual species. We replaced 'species composition' with 'vegetation composition'.*

17. B1-B4: Unit of carbon fluxes in supplementary figures. Per m2?

*Thank you for pointing this out, we changed it accordingly and updated the figures.*

*Response to Referee #3*

18. Thank you for inviting me to review paper "Examining the sensitivity of the terrestrial carbon cycle to the expression of El Nino" by Teckentrup et al. First, may I apologise for taking longer than the expected four weeks to return the review. I realise it can be unfair on the authors to have the Comments section closed, and then another further review appears. For that reason, I have tried to make the review a "light touch", and predominantly suggestions for better framing of the analysis in the future work part. Possibly the most refreshing feature of this paper is that it actually has the confidence to present a "negative result". That is, for the processes investigated by factorial methods, these are likely to have a size that is relatively small compared to the overall impacts of on-going background climate change caused by fossil fuel burning. That is, though, still really important to know, and it does not diminish from the paper. However, by presenting the findings as unimportant also feels like a disservice to the paper findings? As so much recent research into the climate system illustrates, the simultaneous interannual variability of Earth System components does reveal much about potential long-term changes under global warming. Indeed the entire Emergent Constraint concept is based on such an approach. Hence, when placed in that context, the quite specific findings of this analysis become particularly important. I would encourage the authors to at least consider talking about this in the Future Directions part of the manuscript. When parts of ENSO are in a particular phase, what does it tell us about the terrestrial carbon store response, should general climate warming be in that state in a persistent way?

*We have thought about these comments carefully and would like to note that we mention the importance of different expressions of El Nino on interannual timescales of the terrestrial carbon cycle. We concluded that we could nuance our conclusions a little to help resolve the reviewer's comment. We therefore adjusted our conclusions*

*'Based on this analysis, we suggest that our model sensitivity would likely be similar to that displayed by the other TRENDY models, although we would anticipate subtle regional differences, particular in the tropics if an alternative DGVM had been used. Especially for EP El Nino events, LPJ-GUESS diverges from the TRENDY ensemble mean that cannot be explained by nutrient limitation and suggests a different sensitivity to the meteorological drivers (see appendix figure B10).'*

*and:*

*'Our results therefore suggest that the impact of different expressions of El Nino on the carbon cycle on long timescales is likely to be small.'*

*to reflect some of this commentary by the reviewer. Lastly, we add in the discussio:*

*'Interconnections between the terrestrial carbon cycle and ENSO have been widely explored (e.g. Zhang et al., 2018; Rödenbeck et al., 2018; Chylek et al., 2018). Whilst we also find key NBP variability on annual to decadal timescales (see fig. 2),*

*particularly in CP years (accumulated NBP = 7.7 PgC), we did not find that this shorter timescale variability translated into sustained trends (1968-2013) in ecosystem fluxes, or shifts in vegetation distributions (see fig. 3).'*

*We are not sure what the reviewer means in their final point about ENSO and the impact of a particular phase. Our focus in on the character of El Nino, not ENSO. Of course, this is not separate but without understanding more of the reviewer's concern it is hard for us to response.*

19. In the "Future Directions", the authors note that a more formal use of multiple DGVMs will help. The paper does not consider future projections, and it would certainly be interesting to see Figure 2d,e,f extended under the CMIP5/6 ensemble, maybe in a follow-on paper.

   *We agree that work that revisits this question for a future climate may well be warranted. Studies indicate changes in the properties of El Niño events, i.e. magnitude (e.g. Wang et al., 2019) as well as spatial features (e.g. Yeh et al., 2009). However, the representation of ENSO diversity in CMIP5 and CMIP6 models is associated with high uncertainty due to model biases especially in the equatorial Pacific, resulting in low model agreement (e.g. Freund et al., 2020). In order to get robust results, a future experiment set-up would need numerous climate forcing input datasets. In addition, we think that a future study would require multiple DGVMs since the results may be very sensitive to assumptions related to vegetation responses to $[CO_2]$ and interactions with nutrients (Zaehle et al., 2014). We therefore viewed this as beyond the scope for this paper.*

   *In order to reflect these comments by the reviewer we have added the following into the future directions text:*

   *'Moreover, exploring the impact of different expressions of El Nino in a future climate would be worthwhile. However, we note that this would probably require multiple DGVMs to account for the uncertainty associated with the vegetation responses to $[CO_2]$ and interactions with nutrients (Zaehle et al., 2014). In addition, the representation of ENSO diversity in CMIP5 and CMIP6 models is highly uncertain due to model biases, especially in the equatorial Pacific, resulting in low model agreement (e.g. Freund et al., 2020). Therefore, to obtain robust results, a future experimental design would also require an ensemble of climate forcing input datasets.'*

20. Assessment of future findings will also have to be related to how well individual ESMs performing in projecting ENSO characteristics. The authors could also provide a couple of sentences on how others might be encouraged by this analysis to use data to assess the carbon cycle components of their analyses. Datasets do exist of the carbon cycle components, and for instance of NPP ("MODIS NPP"?). While some gridded datasets of terrestrial carbon do contain aspects of models in them e.g. to disaggregate from point to all locations, they still remain highly useful guides and are still "measurements" as such. What would comparisons show between the model-based findings of this paper and terrestrial carbon cycling measurements?

*We agree with the reviewer that it is useful to assess how well LPJ-GUESS simulates the terrestrial carbon cycle. We note that LPJ-GUESS is a well-established DGVM that has been evaluated against observations in previous studies (e.g. Smith et al., 2014). A previous study (Wang et al., 2018) found that the TRENDY models generally captured the anomalies in the terrestrial carbon cycle associated with different expressions of El Nino. We showed in the appendix figure B10 (see below fig.1) that LPJ-GUESS lies within the uncertainty range of the TRENDY ensemble.*

[Figure]

*Fig. 1. Monthly composite anomalies during the El Nino developing (y0) and decaying (y1) year in gross primary production (GPP; green lines) and terrestrial ecosystem respiration (TER; sum of autotrophic and heterotrophic respiration; red lines) for all CP and EP El Nino events listed in appendix table A1 averaged over the globe, the tropics (23°S–23°N) and Australia. The dotted lines show the TRENDY v7 composite, the solid lines are the individual LPJ-GUESS run (compare Wang et al., 2018). The shaded area shows the model spread of the individual TRENDY models.*

*The spatial distribution of the summed composite GPP anomalies (see fig. 2) further shows that LPJ-GUESS picks up the main features of anomalies associated with EP El Nino events (see fig. compare TRENDY composite and individual models). The anomalies in GPP associated with CP El Nino events however display generally weaker responses in Brazil and Western Africa compared to the ensemble mean and most individual models. This low sensitivity might also explain the relatively low correlation and $R^2$ values in figure 1 for tropical regions and may dampen the overall response to the CP only scenario. We note however that LPJ-GUESS is still within the model range and can therefore be viewed as representative. In addition, LPJ-GUESS has a strong negative bias in Australia. As our results show, Australia does not make a large contribution to long-term*

changes in any of the carbon fluxes and pools. We therefore conclude that LPJ-GUESS was suitable to address our experiment.

We now include figure 2 in the manuscript and add in the future directions:

'The spatial distribution of the composite anomalies shows that LPJ-GUESS captures the features of anomalies in GPP associated with EP El Nino events compared to the individual models and the TRENDY model ensemble (see appendix figure B11). In contrast, LPJ-GUESS generally simulates weaker anomalies in GPP associated with CP El Nino events in Brazil and Western Africa compared to the ensemble mean and most individual models. This low sensitivity might also explain the relatively low correlation and R2 values in appendix figure B10 for tropical regions and may dampen the overall response to the CP-only-scenario. We note however that LPJ-GUESS still is within the model range and can therefore be viewed as representative. In addition, LPJ-GUESS has a strong negative bias in Australia. As our results show, Australia does not make a large contribution to long-term changes in any of the carbon fluxes and pools.'

[Figure]

Fig. 2: Composite anomalies in gross primary production (GPP) summed over the the El Nino developing and decaying year for all CP and EP El Niño events listed in tab. B1 for the individual TRENDY models, the TRENDY composite and the individual LPJ–GUESS run (compare Wang et al., 2018).

*Finally, we argue that a comparison with satellite derived observations can only be helpful to a limited extent. In our study, we focussed on the overall, long-term response of NBP to perturbations in El Nino. Future work that wished to probe seasonal and sub-seasonal responses may be able to exploit satellite-derived datasets (e.g. leaf area index (Zhu et al., 2013); GRACE terrestrial water storage (Rodell et al., 2004)) to examine the sensitivity of modelled carbon and water fluxes in response to CP and EP El Nino events. Further, as the reviewer already mentions, satellite derived GPP or NPP products are based on light-use efficiency models themselves and therefore are not directly observed. We therefore include in the future directions*

*'Lastly, a comparison with satellite-derived observations might help to estimate whether LPJ-GUESS or indeed an alternative DGVM, captures the correct sensitivity in the response of vegetation dynamics to ENSO events. Nevertheless, as direct global measurements of carbon fluxes do not exist, and those that do are often based on models themselves, future work might restrict comparison to less direct proxies of variability e.g. leaf area index (Zhu et al., 2013) and/or GRACE terrestrial water storage (Rodell et al., 2004).*

21. The authors could then discuss in a short paragraph how data can constrain which aspects of land surface responses are performing well, and where there are deficiencies. Once constrained, the implications under future climates can be characterised. Although ecosystem acclimation effects might have to be accounted for, this would still offer an extra way to use current interannual variability to tell us about climate impacts. That is the variations might tell us terrestrial carbon cycle response under a permanently adjusted near-surface climatic state.

*Our results point to an overall lack of sensitivity of the simulated carbon cycle over the longer term. Given the sensitivity is small, constraining elements of the response would most likely lead to an even smaller response. This would not affect our conclusions therefore. Nevertheless, in our revision we do note an important point of difference in the sensitivity of LPJ-GUESS vs other DGVMs:*

*'We also examined the sensitivity of our results to the use of a nitrogen cycle with LPJ-GUESS (see appendix figure B10) but did not find a strong sensitivity, most likely because nitrogen is not thought to be strongly limiting in the tropics (Vitousek et al., 1984). Based on this analysis, we suggest that our model sensitivity would likely be similar to that displayed by the other TRENDY models, although we would anticipate subtle regional differences, particular in the tropics if an alternative DGVM had been used. Especially for EP El Nino events, LPJ-GUESS diverges from the TRENDY ensemble mean that cannot be explained by nutrient limitation and suggests a different sensitivity to the meteorological drivers (see appendix figure B10).'*

22. This paper provides a framework of which ENSO" expressions" to focus on, on the path to constraining future projections of land carbon cycle change. The paper includes a particularly good introduction, and the broad literature search is undertaken well, capturing all the main recent papers on ENSO-Carbon cycle

teleconnections. I am happy to see any new paper version, and I will try and return any further comments much more promptly.

*Thanks for your positive comments.*

Small things

23. The word "expression" is used quite a bit e.g. in the discussion of the Central-Pacific and Eastern-Pacific features of El Nino. "Attributes" or "features" may be better words?

    *The word "expression" is different from "attributes" and "features". We think it is probably the best word to use in this context, and it is a word others have used (for example, Tippett et al., 2020) in this context.*

24. Can the diagrams could be tidied up a little more? To my eyes at least, some of the features of – for instance – Figure 2 are difficult to see. Slightly thicker curve linewidths might help, and without obscuring each other.

    *We thank the reviewer for the suggestion and updated the figures accordingly.*

25. A better use of the colourbars would help in Figure B1 for instance, to understand better the geographical spread. To achieve this could be by including colour steps that are not all of identical amounts. Clustering of some colour bounds more around the zero value will reveal more information in the maps?

    *We thank the reviewer for the suggestion. The main point of our maps is to show that regional changes in the carbon fluxes and pools are small as well to further support that changes in the analysed variables might not be significant. We argue that a difference of -50 – 50 PgC for cumulative NBP over 45 years or for carbon pools is minor. Therefore, a more detailed representation would not lead to different conclusions.*

References

Capotondi, A., Wittenberg, A. T., Newman, M., Di Lorenzo, E., Yu, J., Braconnot, P., Cole, J., Dewitte, B., Giese, B., Guilyardi, E., Jin, F., Karnauskas, K., Kirtman, B., Lee, T., Schneider, N., Xue, Y., & Yeh, S. (2015). Understanding ENSO Diversity, *Bulletin of the American Meteorological Society*, *96*(6), 921-938. Retrieved Dec 17, 2020, from https://journals.ametsoc.org/view/journals/bams/96/6/bams-d-13-00117.1.xml

Chylek, P., Tans, P., Christy, J., Dubey, M.. (2017). The carbon cycle response to two El Nino types: An observational study. Environmental Research Letters. 13. 10.1088/1748-9326/aa9c5b.

Freund, M., Brown, J., Henley, B., Karoly, D. and Brown, J.. (2020). Warming patterns affect El Niño diversity in CMIP5 and CMIP6 models. Journal of Climate. 10.1175/JCLI-D-19-0890.1.

Park, S.-W., Kim, J.-S., Kug, J.-S., Stuecker, M. F., Kim, I.-W., & Williams, M. (2020). Two aspects of decadal ENSO variability modulating the long-term global carbon cycle. *Geophysical Research Letters*, 47, e2019GL086390. https://doi.org/10.1029/2019GL086390

Rodell M, P. R. Houser, U. Jambor, J. Gottschalck, K. Mitchell, C.-J. Meng, K. Arsenault, B. Cosgrove, J. Radakovich, M. Bosilovich, J. K. Entin, J. P. Walker, D. Lohmann, and D. Toll (2004) : The Global Land Data Assimilation System. Bulletin of the American Meteorological Society, vol 85 (3), pp 381-394.

Smith, B., Wårlind, D., Arneth, A., Hickler, T., Leadley, P., Siltberg, J., and Zaehle, S.: Implications of incorporating N cycling and N limitations on primary production in an individual-based dynamic vegetation model, Biogeosciences, 11, 2027–2054, https://doi.org/10.5194/bg-11-2027-2014, 2014

Tippett, M.K., L'Heureux, M.L. Low-dimensional representations of Niño 3.4 evolution and the spring persistence barrier. *npj Clim Atmos Sci* **3,** 24 (2020). https://doi.org/10.1038/s41612-020-0128-y

Vitousek, P. M. Litterfall, nutrient cycling, and nutrient limitation in tropical forests. Ecology 65, 285–298 (1984).

Wang, J., Zeng, N., Wang, M., Jiang, F., Chen, J., Friedlinstein, P., Jain, A. K., Jian, Z., Ju, W., Lienert, S., Nabel, J., Sitch, S., Viovy, N., Wang,H., and Wiltshire, A. J.: Contrasting interannual atmospheric CO2 variabilities and their terrestrial mechanisms for two types of El Niños, Atmospheric Chemistry and Physics, 18, 10 333–10 345, https://doi.org/10.5194/acp-18-10333-2018, https://www.atmos-chem-phys.net/18/10333/2018/, 2018

Wang, B., Luo, X., Yang, Y.-M., Sun, W., Cane, M., Cai, W., Yeh, S.-W. and Liu, J.. (2019). Historical change of El Niño properties sheds light on future changes of extreme El Niño. Proceedings of the National Academy of Sciences. 116. 201911130. 10.1073/pnas.1911130116.

Yeh, S.-W., Kug, J.-S., Dewitte, B., Kwon, M., Kirtman, B. and Jin, F.. (2009). El Nino in a changing climate.

Yu, J.-Y. and Kim, S. T.: Identifying the types of major El Niño events since 1870, International Journal of Climatology, 33, 2105 2112,465https://doi.org/10.1002/joc.3575, https://rmets.onlinelibrary.wiley.com/doi/abs/10.1002/joc.3575, 2013.

Zaehle, S., Medlyn, B.E., De Kauwe, M.G., Walker, A.P., Dietze, M.C., Hickler, T., Luo, Y., Wang, Y.-P., El-Masri, B., Thornton, P., Jain, A., Wang, S., Warlind, D., Weng, E., Parton, W., Iversen, C.M., Gallet-Budynek, A., McCarthy, H., Finzi, A., Hanson, P.J., Prentice, I.C., Oren, R. and Norby, R.J. (2014), Evaluation of 11 terrestrial carbon–nitrogen cycle models against observations from two temperate Free-Air $CO_2$ Enrichment studies. New Phytol, 202: 803-822. doi:10.1111/nph.12697

Zhu, Z.; Bi, J.; Pan, Y.; Ganguly, S.; Anav, A.; Xu, L.; Samanta, A.; Piao, S.; Nemani, R.R.; Myneni, R.B. Global Data Sets of Vegetation Leaf Area Index (LAI)3g and Fraction of Photosynthetically Active Radiation (FPAR)3g Derived from Global Inventory Modeling and Mapping Studies (GIMMS) Normalized Difference Vegetation Index (NDVI3g) for the Period 1981 to 2011. *Remote Sens.* **2013**, *5*, 927-948.